# Radar and ground-level measurements of clouds and precipitation collected during the POPE 2020 campaign at Princess Elisabeth Antarctica

Alfonso Ferrone[1, 2] and Alexis Berne[1]

[1]Environmental Remote Sensing Laboratory, École Polytechnique Fédérale de Lausanne (EPFL), Lausanne, Switzerland
[2]MeteoSwiss, via ai Monti 146, Locarno, Switzerland

**Correspondence:** Alexis Berne (alexis.berne@epfl.ch)

**Abstract.** The datasets presented in this article were collected during a four-months measurement campaign at the Belgian research base Princess Elisabeth Antarctica (PEA). The campaign, named PEA Orographic Precipitation Experiment (POPE), was conducted by the Environmental Remote Sensing Laboratory of the École Polytechnique Fédérale de Lausanne, with the logistical support of the International Polar Foundation, between the end of November 2019 and the beginning of February 2020. The datasets have been collected at five different sites. A W-band Doppler cloud profiler and a Multi-Angle Snowflake Camera (MASC) have been deployed in the immediate proximity of the main building of the station. An X-band dual-polarization Doppler scanning weather radar was installed 1.9 km south-east of PEA. Information on the various hydrometeor types have been derived from its measurements, as well as from the images collected by the MASC. The remaining three sites were located in a transect across the mountain chain south of the base, between 7 and 17 km apart from each other. At each site, a K-band Doppler profiler and an automated weather station have been deployed. A pyrgeometer and a pyranometer accompanied the instruments at the site in the middle of the transect. A case study, covering the precipitation event recorded on 23 December 2019, is presented to illustrate the various datasets. Overall, the availability of radar measurements over a complex terrain, relatively far from a scientific base, is extremely rare in the Antarctic context, and opens a wide range of possibilities for precipitation studies over the region.

## 1 Introduction

Princess Elisabeth Antarctica (PEA) is a polar research base owned by the Belgian government and managed by the International Polar Foundation (IPF), located at a latitude of 71° 56' 59.64" S and a longitude of 23° 20' 49.56" E. The station has been built on the Utsteinen Nunatak in the immediate vicinity of the Sør Rondane Mountains, a mountain chain located in the Queen Maud Land region of East Antarctica. While the altitude of PEA is 1382 m above the mean sea level (a.m.s.l.), the tallest peaks in a 30 km radius from it reach up to up to 3000 m. With its distance of approximately 180 km from the coast, the base is also one of the few inland scientific facilities on the Antarctic continent. Its period of operation is limited to the sole austral summer, even though few permanent scientific observatories have been installed in its close vicinity and are able to collect measurements all year round.

The surface mass balance (SMB) of the Queen Maud Land has been studied, as well as the influence of precipitation. In general, the importance of the latter lies in its role as the main input to the SMB of the Antarctic ice sheet (Van Wessem et al., 2018). Given the extension of the continent and the inner variability in meteorological and climate conditions, differences can be expected between the amount and frequency of precipitation in each location. In-situ measurements (Braaten, 2000; Fujita et al., 2011) and simulation results (Schlosser et al., 2010) suggest that in Queen Maud land a few substantial events are the main contributors to the total snowfall accumulation. Given the aforementioned role of precipitation, these events are also the major contributors to the local SMB (Boening et al., 2012), and Medley et al. (2018) suggests that the recent increase in precipitation over the region may partially mitigate the local loss of ice mass from the continent.

Focusing the analysis to PEA, local measurements confirm the relevance of these few significant events. The precipitation events themselves have been characterized by Gorodetskaya et al. (2013), identifying the strong easterly winds, high specific humidity, increased temperature, low atmospheric pressure, and the vicinity to a cyclonic system as the typical conditions under which they occur. A later study by Gorodetskaya et al. (2014) investigated the role of atmospheric rivers and their poleward transport of moisture, identifying this mechanism as the one responsible for the largest part of the accumulation at PEA between May 2009 and February 2011.

Measurements at PEA are not limited to the ground-level meteorological variables used in the two studies mentioned above. Since 2010, a Micro Rain Radar 2 (MRR-2), a profiling K-band (24 GHz) Doppler weather radar manufactured by Metek (Klugmann et al., 1996), has been installed on the rooftop of PEA, alongside a ceilometer and an upward-looking infrared pyrometer. This installation, as part of the HYDRANT observatory, is described by Gorodetskaya et al. (2015), who provides also an example of the possible applications of the collected datasets: the first characterization of clouds and precipitation at the site. A later study by Durán-Alarcón et al. (2019) used the same dataset to continue the characterization of snowfall at the site, focusing on the distinction between precipitation that sublimates before reaching the surface (virga) and the one that instead arrives at the ground.

The same radar measurements allowed a variety of scientific investigations on different topics. For instance, Souverijns et al. (2018b) used them to evaluated satellite-based radar products. A similar comparison, focusing on the snowfall rate, is presented by Lemonnier et al. (2019). The availability of the MRR-2 at the ground offers also an opportunity to focus on some limitations of satellite measurements, as shown by Maahn et al. (2014) in regard to the radar blind zone above the surface. Additional topics investigated thanks to the MRR-2 datasets collected at PEA include the study of the relationship between the radar reflectivity factor and the snowfall rate at the site (Souverijns et al., 2017), and the local contribution of precipitation to the surface mass balance (Souverijns et al., 2018a).

The relatively high number of studies that were enabled by the availability of the HYDRANT dataset underlines the importance of radar measurements in inland Antarctica. A major contribution to its importance is its multi-year time span, which makes it possible to capture changes even on the annual scale. However, the fixed location at which the instrument has been installed does not permit an analysis of the spatial variation in snowfall over the surroundings of the base. In an attempt to address this scientific questions, the Environmental Remote Sensing Laboratory of the École Polytechnique Fédérale de Lausanne (EPFL-LTE), with the logistical support of the IPF, designed and conducted a measurement campaign taking place at PEA and

in its vicinity. The project has been named PEA Orographic Precipitation Experiment (POPE), underlying the importance of monitoring snowfall over the complex terrain directly south of the base, and the resulting variability. The field campaign took place between 23 October 2019 and 11 February 2020, even though the period of data collection varies between the various scientific instruments that took part in the campaign. The instrument set includes remote-sensing ones, i.e. weather and cloud radars, and instruments providing information at the ground level, such as Automated Weather Stations (AWS), a Multi-Angle Snowflake Camera (MASC) and radiometers. Part of this instrumentation has been deployed in a transect across the mountains directly south of PEA, reaching a maximum distance of 30 km from the base and representing one of the major novelties of this campaign.

This article presents the data collected during the POPE measurement campaign. Section 2 describes the sites, the scientific instruments deployed at each of them and the data they collected. Section 3 details the processing of the measurements. Section 4 presents a brief analysis of a precipitation event, illustrating some of the measurements and the derived quantities available in the datasets. Finally, Section 5 concludes the article.

## 2 Measurement sites and instruments

The characterization of the spatial variability of snowfall over the complex terrain surrounding PEA is the scientific question motivating the POPE campaign. To capture this spatial variability, instruments have been deployed at different sites. Their locations are illustrated in Figure 1 by a series of markers overlaid to the digital elevation model (Howat et al., 2019) of the region. Each of these sites and the associated measurements will be presented in the following subsections.

### 2.1 Instruments at the base

Two scientific instruments have been installed in the immediate vicinity of PEA. This choice was mostly dictated by logistical constraints, related either to a high power consumption or the need for a frequent access to the measuring device.

The W-band Doppler cloud profiler and the Multi-Angle Snowflake Camera (MASC), were able to collect valid measurements for the majority of the deployment period. It should be mentioned that a tropospheric lidar and a weighing precipitation gauge were also deployed, but suffered from technical problems and unfortunately no valid data was recorded during the campaign. These two instruments will not be further discussed in the remaining sections of this article.

#### 2.1.1 W-band Doppler cloud profiler

A vertically-pointing W-band (94 GHz) Doppler cloud radar (Küchler et al., 2017), hereafter referred to as WProf, was deployed on a wooden pallet fixed to ice-free rocks of the ridge south of PEA. This radar, officially known by the commercial name of RPG-FMCW-94-SP, is developed by Radiometer Physics GmbH (RPG). This instrument has been collecting measurements between 27 November 2019 and 6 February 2020. However, the first days of deployment saw multiple changes in the configuration of the instrument and have been excluded from the dataset, whose initial date is therefore moved to the 3 December 2019, at 17:00 UTC. The time at which the instrument was collecting measurements is shown in Figure 2. A

metal shipping container, used as shelter for the data acquisition systems by several scientific experiments, is present on the same ridge, approximately 10 meters to the north-east of the radar. The vertical pointing of the radar was checked using two perpendicular levels placed on the instrument.

This instrument allows a large degree of flexibility when deciding the configuration used for the data acquisition. In our case, the final decision was a three-fold split of the chirp table, which divides the vertical extent into three consecutive sections. This particular split allows a minimum detectable reflectivity factor below -15 dBZ throughout the profile. As sensitivity decreases with height, the transition to the region of the profile associated with the next chirp brings a sharp increase in sensitivity, while the vertical resolution and the Nyquist velocity interval decrease. Therefore, the measurements are characterized by a relatively high vertical resolution in the lowest range gates, and the resolution is progressively lowered to keep a sufficiently high sensitivity for the whole profile. The decrease in Nyquist velocity in the second and third chirp is consistent with the lower vertical velocity expected in the upper regions of the profile. The parameters chosen for the period from 3 December 2019 at 17:00 UTC onward are detailed in Table 1.

### 2.1.2 Multi-Angle Snowflake Camera

The MASC provides high-resolution pictures of ice hydrometeors falling through its detection area, as well as an estimate of their vertical velocity. At its core are three cameras, laying on the same plane and pointing, with an angular separation of $36°$, towards a common focal point at about 10 cm distance from each of them. This focal point is in the middle of a metal ring, on which a system of infrared emitters at different heights triggers the cameras and provides the information needed for the fall speed computation (Garrett et al., 2012).

The MASC was deployed on the rooftop of the central PEA building. The instrument was installed on a metal pole, fixed to the railings of the base, in the immediate proximity of the permanent ceilometer installation, part of the HYDRANT observatory described by Gorodetskaya et al. (2013). The MASC was operational between 28 November 2019 and 05 February 2020, but a problem in the power supply resulted in a lack of measurements for the period from 10 to 17 January 2020. During its operational period, the instrument captured 17353 image triplets. The data collected by the MASC and some derived quantity, such as the classification of the hydrometeor types (Praz et al., 2017) and the detection of blowing snow (Schaer et al., 2020), are accessible as part of the MASCDB library (Grazioli et al., 2022) and are therefore not been included in the dataset associated with the present article.

### 2.2 Transect of K-band Doppler profilers

Three Micro Rain Radars PRO (MRR-PRO), a model of K-band Doppler profilers manufactured by Metek, have been deployed over a transect across the mountains directly south of PEA. Their small size and low power consumption made them nicely suitable for deployment in remote sites, with limited access to maintenance and no connection to the power grid of PEA. Each of the three radars has been installed on a vertical pipe, inserted in the ice directly below it, and stabilized using guy-wires connected to four anchor points at different angles, to avoid excessive vibration of the instrument. The vertical pointing of the instrument was checked using the circular level included in the radars.

The transect comprises the three different locations shown in Figure 1, approximately between 7 and 17 km apart from each other, and a maximum distance from PEA of 30 km. To distinguish the three sites we will use the serial number of the radar deployed at each of them:

- MRR-PRO 06, deployed close to one of the mountain ridges (latitude 72° 7' 4.8" S, longitude 23° 20' 49.2" E). Its position has been chosen as a compromise between the need of getting as close as possible to the mountain peaks and the accessibility constraints of the region. The chosen location is at an altitude of 2000 m a.m.s.l.

- MRR-PRO 22, deployed at the beginning of the plateau (latitude 72° 13' 37.2" S, longitude 23° 11' 27.6" E). This location is the highest of the three (2360 m a.m.s.l.). Given its position relative to the nearby orography, it can also be generally considered downstream of the mountain top in the typical atmospheric flow during significant precipitation events.

- MRR-PRO 23, deployed in a valley connecting the plateau to the lower plains (latitude 72° 6' 50.4" S, longitude 23° 30' 50.4" E). This is the lowest of the three locations (1543 m a.m.s.l.), just about 150 m above PEA (1382 m a.m.s.l.). It is also East of the main peaks, implying an upstream position with respect to the mountain.

The precise choice of the three locations is further explained in Appendix A. The duration of each collection period varies slightly among the three profilers: 11 December 2019 to 27 January 2020 for the MRR-PRO 06, 14 December 2019 to 28 January 2020 for the MRR-PRO 22, 12 December 2019 to 28 January 2020 for the MRR-PRO 23. The time at which the three profilers have collected measurements is shown in Figure 2, where it can be compared with the time series associated with the other radars.

### 2.2.1 Automated weather stations and radiometers

An automated weather station, manufactured by Vaisala, has been has been collocated to each MRR-PRO, at a height of approximately 1.5 m above the ground. Two different models of weather stations have been used: the models WXT536, co-located with the MRR-PRO 22 and MRR-PRO 23, and the WXT520, at the MRR-PRO 06 site. The weather stations use the same measurement setup, their data have been collected at the same temporal resolution, and we do not expect any significant discrepancy in their observations because of the difference in the model. The values of wind direction and speed, atmospheric pressure, air temperature, and relative humidity with respect to liquid water have been collected ensuring time synchronization with the radar measurements.

Additionally, a pyranometer (manufactured by Kipp & Zonen, model CMP3) and a pyrgeometer (Kipp & Zonen, CGR3) have been deployed alongside the MRR-PRO 06, on an horizontal pipe connecting the radar to the AWS. The two radiometers provide estimates of the total downwelling irradiance, respectively in the shortwave (wavelength range between 300 nm and 2800 nm) and the longwave (4500 nm to 42000 nm).

## 2.3 X-band dual-polarization Doppler scanning weather radar

The last site of the measurement campaign was approximately 1.9 km to the south-east of PEA, a few hundreds meter east of the Utsteinen Nunatak, as visible in Figure 1. At this location we deployed an X-band (9.41 GHz) scanning Doppler dual-polarization weather radar (MXPol), equipped with a magnetron in its transmission chain, developed by Prosensing (Schneebeli et al., 2013).

The scanning capabilities of this instrument allow the radar to monitor the surrounding area, collecting data along the direction in which the antenna moves. The types of scans performed during this campaign can be divided into two categories: Range-Height Indicator (RHI), during which the radar azimuth stays constant, while the elevation is changed, and Plan Position Indicator (PPI), in which the elevation is kept fixed while the antenna rotates in azimuth. Typically, a series of PPI and RHI is repeated in a fixed order, defining the scanning cycle for the campaign, which was repeated indefinitely during operation.

The radar has been collecting measurements discontinuously for the period between 16 December 2019 and 5 February 2020, being usually manually activated before precipitation events. The position of the radar has been chosen as a trade-off between ease of access, needed to operate it on a regular basis, and visibility on the mountain range. Since the only major obstacle in its vicinity was the Nunatak on the west, MXPol had a clear view on the base, most of the southern peaks and on the North-East, direction of the typical flow associated with precipitation systems. These conditions on the visibility led us to define different types of scan cycles.

The first of them was dedicated to the monitoring of widespread precipitation events, covering a large part of the area around the radar. The duration of this cycle is approximately 6 minutes. Its standard version, in use for the majority of the deployment period, comprises:

- an RHI at 165.7° of azimuth, directed towards the location of the MRR-PRO 23;

- an RHI at 190.1° of azimuth, conducted along an imaginary line passing in between the MRR-PRO 06 and MRR-PRO 22;

- an RHI at 219.1° of azimuth, pointing towards the Vikinghogda, one of the main mountain peaks in the surroundings;

- an RHI at 318.1° of azimuth, passing above PEA;

- an RHI at 39.1° of azimuth, approximately along the NE and completing the hemispherical trajectory initiated by the RHI scan at 219.1°;

- a PPI at 4.4° of elevation, chosen to avoid excessive clutter contamination from the mountains;

- a PPI at 88.9° of elevation, acting as nearly-vertical profile and providing measurements for the calibration of $Z_{DR}$.

The first three RHI scans have been performed in in Fast Fourier Transform (FFT) mode, and the elevation span has been limited between 0.9° and 68.9° to reduce the execution time. The remaining ones, instead, are performed in Dual-Pulse Pair (DPP) mode and terminate at higher elevations, respectively equal to 88.9° and 108.9°. From the PPI scan at 4.4°, we excluded the azimuth sector between 244.4° and 314.7°, to remove the influence of the Nunatak nearby.

The second cycle instead has been designed to monitor the small-scale precipitation events visible directly above some of the
mountain peaks. The duration of this cycle is slightly shorter than the previous one, being approximately equal to 4 minutes. In
addition to the previously discussed RHI scans toward the MRR-PRO sites (165.7° and 190.1° of azimuth), the cycle includes:

- an RHI at 134.0° of azimuth, pointing towards a mountain peak, to the east of the mountain peak Verheyefjellet, over
  which isolated clouds and precipitation were sometimes visible;

- an RHI at 141.3° of azimuth, directed at another peak, with a similar location and set of characteristics as the previous
  one;

- a sector scan (akin to a PPI, but limited in the azimuth extent) at 4.4° of elevation, covering between 127.1° and 234.8°
  of azimuth, providing visibility over most of the mountain range.

These scans have been performed in FFT mode. All the angles presented in the two cycles have been corrected for the azimuth
and elevation offset of the antenna. The determination of the two biases is described in section 3.1.1. The different directions of
the RHI scans are displayed as lines departing radially from the MXPol site in Figure 1, with color and style used to distinguish
between the two cycles.

A temporary version of the scan cycle, containing scans from both the first and second ones, has been used before the 22
December 2019. Since this scan has been executed only 15 times, it has not been included in Figure 1. Its total duration was
considerably longer than the first and second cycles, equal approximately to 11 minutes. This cycle was defined before an
accurate estimate of the azimuth and elevation offset of the antenna was available. Therefore, the direction of each scan was
based solely on a visual estimate of the alignment of the antenna with the target, and the resulting angle differ slightly from the
ones presented before. The following list provides the scans included in the temporary cycle:

- an RHI at 145.1° of azimuth, in place of the one at 141.3° in the second cycle;

- an RHI at 174.1° of azimuth, pointing approximately in the direction of the valley in which the MRR-PRO 23 was
  deployed, even though the direction is not as accurate as the one in the first cycle;

- an RHI at 180.6° of azimuth, in the direction of the mountain peak between the MRR-PRO 06 and MRR-PRO 23. To
  reduce the execution time of the cycle, this direction has been excluded from the first and second cycle;

- an RHI at 190.6° of azimuth, in place of the one at 190.1° in the first cycle;

- an RHI at 219.6° of azimuth, in place of the one at 219.1° in the first cycle;

- an RHI at 320.6° of azimuth, in place of the one at 318.1° in the first cycle;

- an RHI at 44.6° of azimuth, in place of the one at 39.10° in the first cycle;

- a PPI at 1.0° of elevation, in DPP mode;

– a PPI at 88.9° of elevation, in FFT mode.

All the RHI scans in this cycle have been performed in FFT mode. The time of operation of the three scan cycles is shown in
Figure 2.

The decision between the DPP and FFT mode has a significant impact on the final products. Firstly, the full Doppler spectrum
is available only for the files generated by FFT scans, which will result in a slower antenna movement and larger file sizes. The
slow execution of FFT scan is the reason behind the long duration of the temporary cycle, and the reason why some of them
have been replaced by DPP ones in the first cycle. The ones recorded in DPP mode, instead, create smaller files, and the scan
instead is typically faster. The type of the scan also affects the Nyquist velocity: for the FFT ones it has been set to 11 m/s, for
the DDP ones to 39 m/s.

With the sole exception of the PPI scan at 88.9° of elevation, measurements are collected up to a distance of 28.8 km from the
radar for DPP scans, and of 27.6 km for the FFT ones, both with a range resolution of 75 m. The nearly-vertical PPI, instead,
has been limited to a maximum range of 4.8 km, with a resolution of 30 m, to reduce the execution time while focusing solely
on the lower region of the atmosphere, more likely to contain meteorological signal.

## 3   Data processing

This section focuses on the meteorological radars, describing the procedure followed in deriving the radar variables from the
raw data and the calibration techniques used to ensure the accuracy of the measurements. The processing of measurements
collected by the AWS and radiometers is limited to the removal of non-numerical characters from the time-series and the
storage in netCDF4 format (Unidata, 2019), and therefore has not been discussed further in this section.

### 3.1   MXPol

The processing of the data collected by MXPol follows closely the steps described by Gehring et al. (2021) for the same radar
during the ICE-POP 2018 campaign in South-Korea. Starting from the raw spectra collected by the instrument, the method of
Hildebrand and Sekhon (1974) is used for isolating the signal from the noise floor. The former contains the information needed
to determine the polarimetric variables, according to the backscatter covariance matrix procedure illustrated by Doviak and
Zrnic (1993, chapter 8). The creation of the data files is performed using the Python library PyART (Helmus and Collis, 2016).

The processing produces the following radar variables:

– $Z_H$ and $Z_V$, the horizontal and vertical reflectivity factors, stored in logarithmic form and having the units dBZ;

– $Z_{DR}$, the differential reflectivity, also stored in logarithmic form and expressed in dB;

– $SNR_H$ and $SNR_v$, the signal-to-noise ratio on the two polarization channels, in dB;

– $V$, the mean Doppler radial velocity, whose units are $\mathrm{ms}^{-1}$;

– $SW$, the spectral width, also expressed in $\mathrm{ms}^{-1}$;

- $\Psi_{DP}$ and $\Phi_{DP}$, respectively the total differential phase shift and the differential phase shift, in degrees;

- $K_{DP}$, specific differential phase on propagation, in $^\circ\,\mathrm{km}^{-1}$, estimated using the method described by Schneebeli et al. (2014);

- $\rho_{hv}$, the co-polar correlation coefficient, dimensionless.

Detailed information on the storage of the radar variables and the Doppler spectra in the dataset is provided in Appendix B1.

### 3.1.1 Correction of the azimuth and elevation offset

Since MXPol is a scanning weather radar, each of its measurements is characterized by an azimuth and elevation angle. The complexity of the deployment on an ice surface, in a rather remote site, makes achieving a perfect alignment of the antenna extremely difficult. Therefore, an offset that compensates for any misalignment must be computed and subtracted from the angle recorded by the radar.

On the 07, 08, 10 and 13 January we performed series of scans covering a small windows of azimuth and elevation angles around the position of the Sun. By comparing the apparent position of the center of the Sun as seen by the radar with its expected position provided by the Python library Astropy (Astropy Collaboration et al., 2018), we can estimate the offset. This approach is loosely based on the method described by Muth et al. (2012). Our estimate of the offsets are: $108.11^\circ$ for the azimuth, $1.05^\circ$ for the elevation.

The validity of the azimuth estimate has been controlled by pointing the antenna towards recognizable mountain peaks on the south of PEA, checking the difference between the angle of each geographical feature with the one recorded by the radar. The average difference recorded among the five targets tested is $0.2^\circ$, within the expected accuracy for a manual pointing.

### 3.1.2 Calibration

The radar underwent a control of the stability of the signal in July 2018, during a short deployment in Remoray (France). The calibration was performed by Palindrome Remote Sensing GmbH, using a radar target simulator. Overall, the information collected during this short deployment show that the reflectivity factor recorded by MXPol is within $1\,\mathrm{dBZ}$ of the real value. No estimate of the magnitude of a potential drift of the calibration since July 2018 is available.

### 3.1.3 Correction of the differential reflectivity bias

Depending on the target and application, the differential reflectivity measurements require to be calibrated with an accuracy that varies between $0.1\,\mathrm{dB}$ and $0.2\,\mathrm{dB}$ (Ryzhkov et al., 2005). To ensure that our measurements satisfy this condition we computed a time-varying offset using the method described by Ferrone and Berne (2021). According to this technique, the offset values are derived from the vertical PPI scans performed by the radar at regular intervals. Their time variability is determined by a fit of the semi-variogram, whose parameters are used for a 1-dimensional ordinary kriging interpolation of the offset of each scan. For this measurement campaign, we used a spherical model for the fit with the following parameters: the nugget is $0.0021\,\mathrm{dB}^2$,

the range is 540 min and the sill is 0.0043 dB$^2$. The final interpolation output is shown in Figure 3, superimposed to the offset computed from each PPI scans.

### 3.1.4 Determination of the hydrometeor types

The polarimetric variables collected during the RHI scans performed by MXPol have been used to identify the dominant hydrometeor type, according to the algorithm of Besic et al. (2016). The centroids used in the procedure are the same ones mentioned by Gehring et al. (2021), computed on past datasets collected by the same radar. The RHI scans have also been used to determine the hydrometeor mixture in the radar volumes, using the method described by Besic et al. (2018).

The same hydrometeor classification method has been applied to measurements collected in another measurement campaign in Antarctica, albeit at a coastal location rather than an inland one (Gehring et al., 2022). An accurate evaluation of the reliability of these estimates cannot be provided for the current dataset. Some discrepancies can be observed between their value and the one determined by the MASC at the ground level, as exemplified in Section 4. The difference in sampling volume and altitude of measurements between the two sensors could be partially responsible for these discrepancy. We therefore suggest to use those hydrometeor information with care.

## 3.2 WProf

The raw Doppler spectra were not saved, only the part above the noise level identified by the manufacturer's algorithm (Küchler et al., 2017) was. From this signal, the following radar variables have been computed: equivalent reflectivity factor ($Z_e$, expressed in the linear units $mm^6 m^{-3}$), signal to noise ratio ($SNR$, in dB), $V$, $SW$, skewness, and kurtosis (in $ms^{-1}$). Additionally, WProf includes an automated weather station, whose measurements have been included in the data files, and an 89 GHz radiometer, which allowed the retrieval of the Liquid Water Path (LWP) and Integrated Water Vapor (IWV) using the algorithm described by Billault-Roux and Berne (2021). A list of the variables stored in the WProf data files is provided in Appendix B2.

### 3.2.1 Correction of the attenuation by atmospheric gasses

The attenuation of atmospheric gasses has been computed following a procedure analogous to the one used by Gehring et al. (2021) for the same radar, recommended also by Ippolito (1986). The profiles of atmospheric pressure, air temperature and humidity recorded by the radiosondes launched from one of the secondary facilities of the base, at few hundred meters from the deployment site of WProf, have been used as input for the algorithm. However, given the low values of attenuation, below 0.1 dB, we decided not to add the correction to the files in the dataset.

## 3.3 MRR-PRO

The internal algorithm of the MRR-PRO designed by Metek for the computation of the radar variables is targeted mostly toward observations of liquid precipitation. Moreover, the measurements are sometimes affected by artefacts that cover a small subset

of the range gates and can hinder the correct interpretation of the meteorological signal. Other issues in the measurements include a drop in the power recorded in the raw Doppler spectra in the lines adjacent the extremes of the Nyquist velocity interval. To address these issues, the MRR-PRO data have been processed by the ERUO library (Ferrone et al., 2022), which acts on the raw Doppler spectra to extract a cleaner set of radar variables. ERUO also performs a simple dealiasing in the Doppler velocity measurements, and increases the sensitivity in the profiles of attenuated equivalent reflectivity factor. The final set of variables computed for the three MRR-PRO is analogous to the WProf one, with the exception of skewness and kurtosis, which have not been included in the dataset, but can still be derived from the Doppler spectra and noise level available in the data files. A list of the variables stored in the MRR-PRO data files is provided in Appendix B3.

## 3.4   Sensitivity of the radars

In Figure 4, we illustrate the 2-dimensional histogram of the attenuated equivalent reflectivity factors recorded by MXPol, during the verical PPI scans, WProf, and the three MRR-PRO. A threshold of 0 dB has been imposed on the signal-to-noise ratio of all the radars. While this condition alone does not guarantee the removal of all non-meteorological returns, it allows us to filter out faint peaks in the Doppler spectra that could have mistakenly been identified as precipitation. The left side of the distributions of the WProf reflectivity factor reaches consistently lower values when compared with MXPol, indicating that the former has a higher sensitivity than the latter.

In Figure 4.a, clusters of high counts can be observed at range values between 1 and 1.5 km. This behavior is caused by artefacts in the radar data that, unlike the MRR-PRO case, could not be filtered out during processing. The artefacts are usually covered by relatively strong precipitation signal (above 10 dBZ), but visible when the meteorological return is faint. While this behavior would significantly limit the usefulness of the dataset for a profiler, the scanning capability of MXPol and the large visibility radius results in a majority of the radar volumes being unaffected during RHI and low-elevation PPI scans.

Figure 4.b illustrates a particularity of the measurement configuration of WProf. Sudden variations can be observed in the minimum of the distributions at the transition between the chirps defined in Table 1.

The 2-dimensional histograms computed for the MRR-PRO 06, 22, and 23 are shown in Figure 4.c, 4.d, and 4.e, respectively. By comparing the three datasets, a significant difference between the range of $Z_{ea}$ values recorded by the MRR-PRO 22 and the ones from the other two MRR-PRO can be noticed. We hypothesize that this difference is caused by miscalibration issues, but we could not directly test the agreement between the three profilers before or after their deployment at PEA.

Since the three MRR-PRO share the same hardware specifications, we expect a similar sensitivity in their measurements. Following the analysis presented in Ferrone et al. (2022), we can use the quantile 0.01 of the empirical $Z_{ea}$ distribution at each range gate to estimate the sensitivity. The difference between the quantile 0.01 of the MRR-PRO 06 distributions and the same quantile in the MRR-PRO 22 ones is 9 dB, and the interquartile range (quantile 0.75 minus quantile 0.25) of this difference is equal to 2 dB. The difference between the quantiles 0.01 of the MRR-PRO 23 and MRR-PRO 22 is 11 dB, and the interquartile range of this difference is also 2 dB. The similarity between the two differences suggests that an offset of 10 dB should be added to the MRR-PRO 22 to compensate for its lower $Z_{ea}$ values. Given the impossibility of comparing the MRR-PRO 22 with an adequate reference set of measures and the approximate nature of the offset estimate provided above, we suggest to use

this offset with care. The 10 dB have not been added to the MRR-PRO 22 measurements provided in the publicly accessible datasets.

## 4  Significant weather events

Since the weighing precipitation gauge was not able to record valid measurements during the campaign, it cannot be used as reference for identifying the start and end date of snowfall events at the base. For this reason, we have decided to use the time-series provided by the three MRR-PRO to identify the frequency of precipitation in our datasets, identifying 10 separated events in which most of the instruments have collected measurements. The detection rate varies between radars, due to their different sensitivity. For instance, WProf is able to detect the faint return of clouds, resulting in a higher number of days in
which meteorological signal is observed. In the following subsection, the precipitation event recorded on the 23 December 2019 is presented to illustrate the variety of products in the datasets described in the present article.

### 4.1  23 December 2019

On 23 December 2019, significant precipitation has been observed by all instruments part to the POPE campaign. Figure 5 shows the time series of reflectivity factor from WProf, MXPol, and the three MRR-PRO.
Thanks to its high sensitivity, WProf provides a clear view of the vertical structure of precipitation above the base, detecting meteorological signal up to 7 km above the surface. Similar features can be observed in the time-series recorded by MXPol, even though the vertical extent is significantly lower. As mentioned in section 3.4, artefacts are visible in the data, appearing as horizontal lines in the range gates between 1 and 1.5 km.

Over the transect, MRR-PRO 06 and MRR-PRO 22 record meteorological signal more continuously than MRR-PRO 23. In
the highest of the three locations, precipitation reaches the lowest range gate more often than for the other two profilers. In particular, virga are present in the MRR-PRO 23 time-series in the early and late stages of the event.

The complementary measurements provided by the three AWS and the radiometers are shown in Figure 6. The difference in air temperature between the sites in the transect is consistent with their elevation difference. The relative humidity values experience more significant fluctuations, especially at the MRR-PRO 23 site. In particular, high humidity values can be observed
for some periods at the beginning and end of the event, following a pattern similar to the aforementioned virga. A similar difference in behavior between sites can be seen also for the wind direction. The north-eastern direction is dominant in the MRR-PRO 06 and MRR-PRO 22 time series. The location of the MRR-PRO 23 within a valley could explain the difference from the other two sites, with sudden oscillations between a northerly and southerly flow approximately aligned to the valley orientation. The relatively low values of shortwave irradiance at the MRR-PRO 06, visible in Figure 6.e, confirm the presence
of clouds for most of the day.

Information on the hydrometeor types can be derived from both the MASC and the RHI scans performed by MXPol. Figure 7 shows such information for the event currently described. Given the variety of scans in the two cycles, we decided to limit the analysis to the sole RHI directed toward the base. The proportion of three hydrometeor types (crystals, aggregates and rimed

particles) has been computed using the algorithm designed by Besic et al. (2018). In order to display the values in an easily readable time-series, these proportions have been averaged over the radar volumes with an horizontal distance lower than 500 m from the base, obtaining a mean profile for each scan. A vertical resolution of 75 m has been used to create the profiles.

Crystals dominate the whole time series, followed by aggregates, especially in the lowest section of the profile. Rimed particle are less frequent, reaching maximum values barely above 10 %, mostly close to the ground level. While MXPol records precipitation during the whole day, the MASC experiences extended period of lack of observations. Small particles are particularly abundant, even though blowing snow has been filtered out from the dataset. Aggregates appear more often during the first phase of the event, with crystals overtaking them in proportion after 06:00 UTC. Although the timing in the occurrence of graupel in the MASC classification and of rimed particles in the MXPol classification is overall consistent, discrepancies can be observed between their respective magnitude and relative importance compared to crystals and aggregates.

## 5 Conclusions

This article presents datasets collected by meteorological radars and ground-level instruments between December 2019 and February 2020 in the surroundings of the Belgian research base Princess Elisabeth Antarctica. The scientific equipment has been deployed and maintained by the EPFL-LTE laboratory, with the logistical support of IPF, in the context of the PEA Orographic Precipitation Experiment (POPE). This project has among its main objectives the characterization of snowfall over the complex terrain around PEA.

The main novelty of POPE is the deployment of three K-band Doppler profilers (MRR-PRO) and automated weather stations in a transect across the mountain range, reaching a distance of 30 km from the base. A pyranometer and a pyrgeometer accompany one of the profilers in the transect. An X-band Doppler dual-polarization scanning radar, a W-band Doppler cloud profiler and a multi-angle snowflake camera have been installed in the vicinity of PEA.

The availability of polarimetric and Doppler radar measurements, accompanied by snowflake pictures collected at the base, make this dataset particularly suited for snowfall microphysics studies. Additionally, the variety in the locations allows investigation on the spatial variability of snowfall. Future studies may use the datasets presented in this article to improve the current understanding of precipitation in this region of the Antarctic continent.

## 6 Data availability

The datasets described in this article are available at: https://doi.org/10.5281/zenodo.7428690 (Ferrone and Berne, 2023c). The Doppler spectra have been uploaded to separate online archives, given the large size of these data files. The ones collected by the three MRR-PRO are available at: https://doi.org/10.5281/zenodo.7507087 (Ferrone and Berne, 2023a). The Doppler spectra collected by WProf and MXPol are hosted by the Academic Output Archive (ACOUA) service provided by EPFL, at the following address: https://doi.org/10.5075/epfl-lte-299685(Ferrone and Berne, 2023b).

## Appendix A: Choice of the MRR-PRO locations based on the output of atmospheric simulations

To determine the exact locations of the three sites, we used the output of a series of atmospheric simulations, made by using the version 4.0 of the Weather Research and Forecasting (WRF) model (Skamarock et al., 2021) and covering the three Austral summers preceding the campaign, from December 2014 to February 2017. The parent domain has a resolution of 27 km and it contains three nested domains, of 9 km, 3 km and 1 km resolution, centered around PEA. This elevation has been derived from the Bedmap2 1-km resolution Antarctic topography (Fretwell et al., 2013), and the model has been run with 69 vertical levels. The ERA5 reanalysis (Hersbach et al., 2020) has been used to define the boundary and initial conditions. Additional information on the nudging of the parent domain and the physical parameters can be found in Vignon et al. (2019), which describes a set-up akin to the one used for the current analysis.

In all three simulations, a local maximum of precipitation can be found on the mountain chain directly south of PEA, while the valley that connects the plateau to the lower elevations is usually characterized by smaller accumulations. Repeating the simulations for the period December 2019 to January 2020 shows a similar pattern, illustrated in Figure A1.

While the final decision on the precise locations of the MRR-PRO sites is mostly determined by logistical constraints given by the limited accessibility of remote areas, we attempted to capture the variability of the accumulation field around the aforementioned mountain group and nearby valley. As can be seen in Figure A1, the MRR-PRO 23 has been deployed in a location that receives significantly less precipitation at the ground when compared with the other two sites in the transect. The MRR-PRO 06 is close to the mountain peak, and at the edge of the local accumulation maximum. The MRR-PRO 22 is also close to this maximum, but on the other side of the orography and, in the north-easterly flow typical of precipitation events in the region, can be considered downstream to the mountain.

## Appendix B: Summary of the variables in the dataset

This section provides a summary of all the relevant variables available in the data files stored in the dataset.

### B1 Variables in the data files of MXPol

The following polarimetric variables are available for all the scans performed by MXPol:

- the horizontal reflectivity factors ($Z_H$), identified in the files by the short name "Zh";

- the vertical reflectivity factors ($Z_V$), identified by the short name "Zv";

- the differential reflectivity ($Z_{DR}$), identified by the short name "Zdr";

- the signal-to-noise ratio on the horizontal polarization channel ($SNR_H$), identified by the short name "SNRh";

- the signal-to-noise ratio on the vertical polarization channel ($SNR_v$), identified by the short name "SNRv";

- the mean Doppler radial velocity ($V$), identified by the short name "RVel";

- the spectral width ($SW$), identified by the short name "Sw";

- the total differential phase shift ($\Psi_{DP}$), identified by the short name "Psidp";

- the differential phase shift ($\Phi_{DP}$), identified by the short name "Phidp";

- the specific differential phase on propagation ($K_{DP}$), identified by the short name "Kdp";

- the co-polar correlation coefficient ($\rho_{hv}$), identified by the short name "Rhohv".

The hydrometeor classification (Besic et al., 2016) and the retrieval of the proportion of each hydrometeor category in the radar volume (Besic et al., 2018) has been applied to all RHI scans, and produces the following variables:

- the dominant hydrometeor type, identified by the short name "hydro",

- the entropy computed by the classification algorithm, which provides an estimate of the confidence on the label assigned to the volume, identified by the short name "hydroclass_entropy";

- the proportion of each hydrometeor type in the volume, stored in the variables "proportion_AG" (aggregates), "proportion_CR" (ice crystals), "proportion_LR" (light rain), "proportion_RP" (rimed particles), "proportion_RN" (rain),
"proportion_VI" (vertically-aligned ice), "proportion_WS" (wet snow), "proportion_MH" (melting hail);

- the entropy computed by the demixing algorithm, identified by the short name "entropy".

The scans performed in FFT mode allow the recording of the Doppler spectra. Given the large storage space needed to save this type of data, the Doppler spectra have been stored in separate files, containing the following variables:

- the power spectrum recorded on the horizontal polarization channel, identified by the short name "sPowH";

- the power spectrum recorded on the vertical polarization channel, identified by the short name "sPowV".

## B2    Variables in the data files of WProf

The following radar variables are available for all the profiles collected by WProf:

- the equivalent reflectivity factor ($Z_e$), identified in the files by the short name "Ze"

- the signal-to-noise ratio ($SNR$), identified in the files by the short name "SnR";

- the mean Doppler radial velocity, identified in the files by the short name "Mean-velocity";

- the spectral width, identified in the files by the short name "Spectral-width";

- the skewness, identified in the files by the short name "Spectral-skewness";

- the kurtosis, identified in the files by the short name "Spectral-kurtosis";

- the noise level at each range gate gate, identified in the files by the short name "Noise_level";

- the noise floor at each range gate gate, identified in the files by the short name "Noise_floor".

The following retrievals have been included in the data files:

- the Integrated Water Vapor (IWV), identified in the files by the short name "Integrated-water-vapor";

- the Liquid Water Path (LWP), identified in the files by the short name "Liquid-water-path".

The Doppler spectra have been stored separately in files containing the following variables:

- the spectral reflectivity recorded in the lowest part of the profile, according to the first chirp (as described in Table 1), identified in the files by the short name "Doppler-spectrum-vert-chirp0";

- the spectral reflectivity recorded in the middle section of the profile, according to the second chirp, identified in the files by the short name "Doppler-spectrum-vert-chirp1";

- the spectral reflectivity recorded in the higest part of the profile, according to the third chirp, identified in the files by the short name "Doppler-spectrum-vert-chirp2".

The following atmospheric variables, recorded by the automated weather station integrated in the radar, have been included in both types of data files:

- the atmospheric pressure, identified in the files by the short name "Barometric-pressure";

- the air temperature, identified in the files by the short name "Environment-temp";

- the relative humidity with respect to liquid water, identified in the files by the short name "Rel-humidity";

- the horizontal wind direction, identified in the files by the short name "Wind-direction";

- the horizontal wind speed, identified in the files by the short name "Wind-speed".

## B3    Variables in the data files of the MRR-PRO

The following variables are available for all the profiles collected by the three MRR-PRO:

- the attenuated equivalent reflectivity factor ($Z_{ea}$), identified in the files by the short name "Zea";

- the mean Doppler radial velocity, identified in the files by the short name "VEL";

- the spectral width, identified in the files by the short name "SW";

- the signal-to-noise ratio, identified in the files by the short name "SNR";

- the noise level computed by ERUO (Ferrone et al., 2022) at each range gate gate, identified in the files by the short name "noise_level";

- the noise floor computed by ERUO at each range gate gate, identified in the files by the short name "noise_floor".

The Doppler spectra have been stored separately in files containing the following variables:

- the raw spectral power, identified in the files by the short name "spectrum_raw"

- the transfer function, used to convert the raw spectral power to spectral reflectivity, as described in Ferrone et al. (2022), and identified in the files by the short name "transfer_function".

## B4   Variables in the data files of the automated weather stations

The variables recorded by the three weather stations are:

- the atmospheric pressure, identified in the files by the short name "pressure";

- the air temperature, identified in the files by the short name "temperature";

- the relative humidity with respect to liquid water, identified in the files by the short name "relative_humidity";

- the horizontal wind direction, identified in the files by the short name "wind_speed";

- the horizontal wind speed, identified in the files by the short name "wind_direction".

## B5   Variables in the data files of the radiometers

The variables recorded by the pyrgeometer and pyranometer are:

- the total downwelling irradiance in the shortwave, identified in the files by the short name "shortwave_irradiance";

- the total downwelling irradiance in the longwave, identified in the files by the short name "longwave_irradiance".

*Author contributions.*   A.F. and A.B. designed the study. A.F. and A.B. deployed and maintained the instruments at PEA. A.F. processed the datasets. A.F. prepared the manuscript with contributions and supervision from A.B.

*Competing interests.*   A.F. declares that no competing interests are present. A.B. is associate editor for AMT.

*Disclaimer.*   This research was funded by the Swiss National Science Foundation (grant number 200020-175700/1) the Swiss Polar Institute (Polar Access Fund 2019 and Exploratory Grant).

*Acknowledgements.* We are grateful to all the EPFL-LTE collaborators for their contributions to the preparation of the measurement campaign and for their help in solving issues with the instruments. In particular, we would like to thank Antoine Wiedmer for developing most of the technical solutions that allowed the remote deployment of the instruments, Anne-Claire Billault–Roux for her contribution in the processing of the measurements collected by MXPol and WProf, and Étienne Vignon for executing the WRF simulations. Our gratitude also goes to all the personnel of the Princess Elisabeth Antarctica base: the staff of the International Polar Foundations, the technicians on site, and the field guides that made the remote deployment of the radar possible.

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

**Table 1.** Parameters of the measurement setup of WProf. Each vertical profile is divided into three regions, denoted by a chirp number, listed in the first column of this table. The second and third column provide the vertical range and the range resolution for each of the chirps. The Nyuist velocity ($v_{ny}$) and velocity resolution of the Doppler spectra collected in each of region of the profiles are listed in the fourth and fifth columns.

| Chirp num. | Vertical range [m] | Vertical res. [m] | $v_{ny}$ [m/s] | Vel. res. [m/s] |
|---|---|---|---|---|
| 1 | 104 - 999 | 7.5 | 10.8 | 0.02 |
| 2 | 1008 - 3496 | 16.2 | 7.0 | 0.01 |
| 3 | 3512 - 8586 | 32.5 | 3.3 | 0.007 |

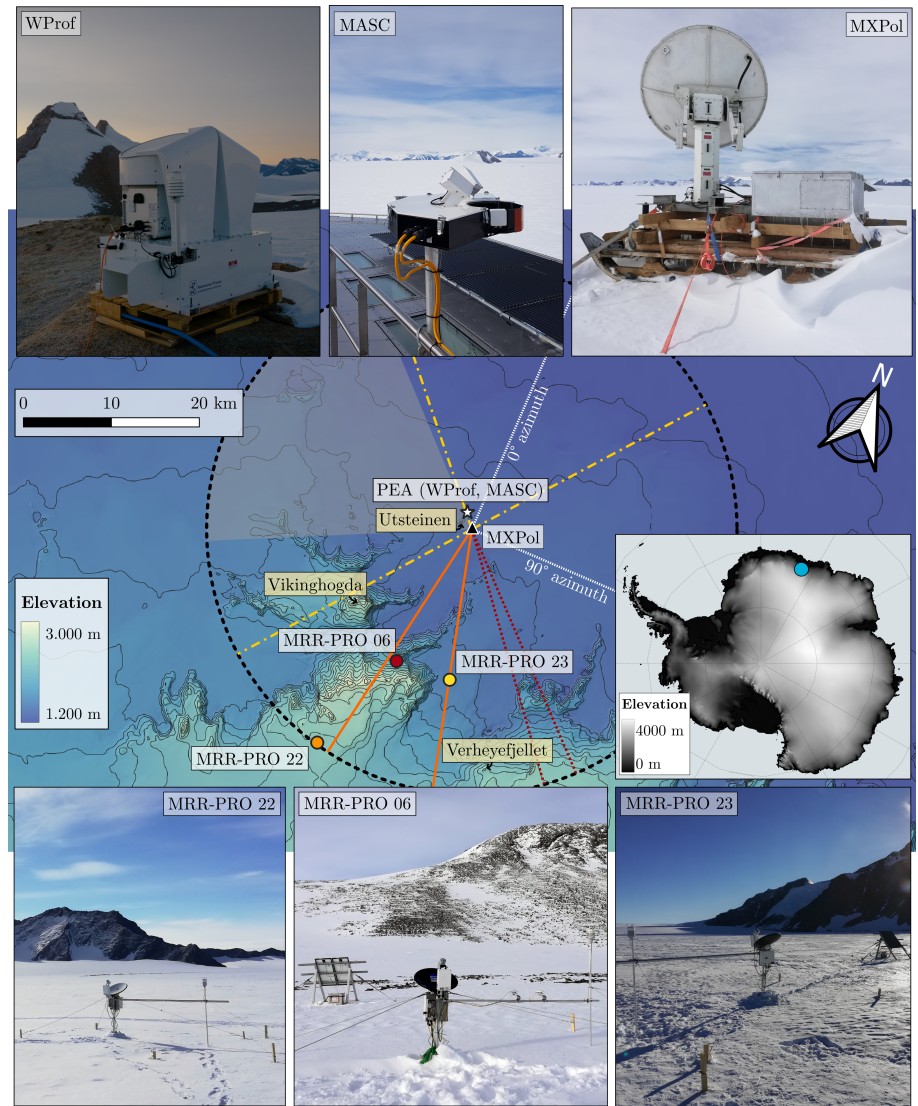

**Figure 1.** Elevation map of the region surrounding PEA. The markers and the accompanying text on white background superimposed to the map highlights the location of the instruments deployed during the campaign. The mountain peaks mentioned in the manuscript are denoted by text on a pale yellow background. A picture of each of them is provided in the six panels above and below the map. The black dashed circle shows the maximum detectability radius of MXPol, while the coloured radii illustrate the direction of the RHI scans: the yellow dash-dotted lines show the scan belonging only to the first cycle, the red dotted ones belong only to the second cycle, and the continuous orange ones belong to both. The shaded gray area shows the range of azimuth angles excluded from the PPI scans. The small panel in the right-hand side of the figure shows the position of PEA on the Antarctic continent. In both panels the elevation has been derived from the REMA digital elevation model (Howat et al., 2019).

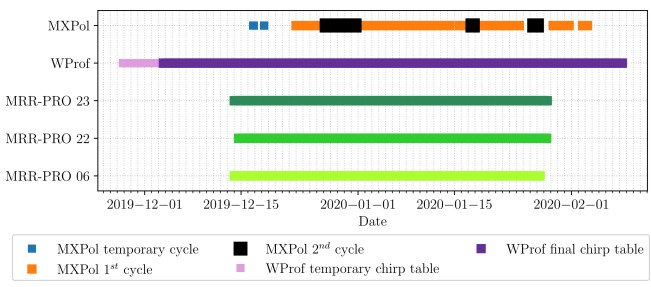

**Figure 2.** Time at which measurements have been collected by the five radars. The time series at the top of the figure is divided according to the scan cycles of MXPol. The blue markers denote the date and time of execution of scans in the temporary cycle, the orange ones the scans in the first cycle, and the black ones the scans in the second cycle. The second time series is divided according to the chirp table used by WProf: the definitive one, used for most of the campaign, is shown in purple, while the temporary ones preceding it are shown in pink. The time series of the three MRR-PRO are shown as three different shades of green, in the three bottom rows of the panel.

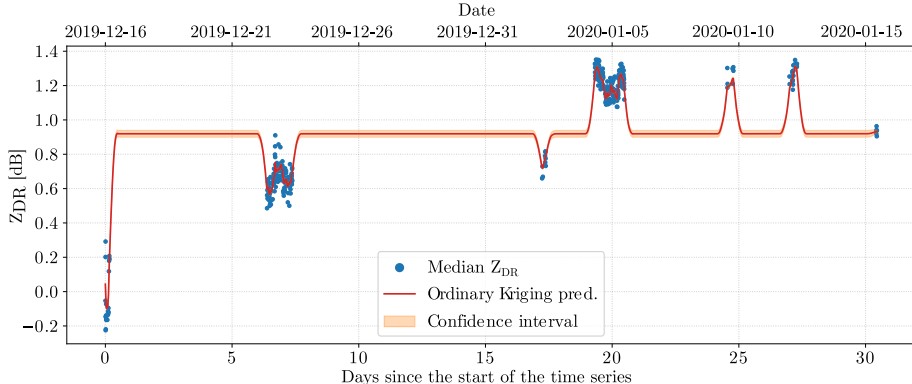

**Figure 3.** Time series of the differential reflectivity bias computed for MXPol. The continuous red line illustrates the final bias provided by the kriging interpolation, its uncertainty interval is highlighted in orange, and the median offset derived from each vertical scan is displayed as blue dots.

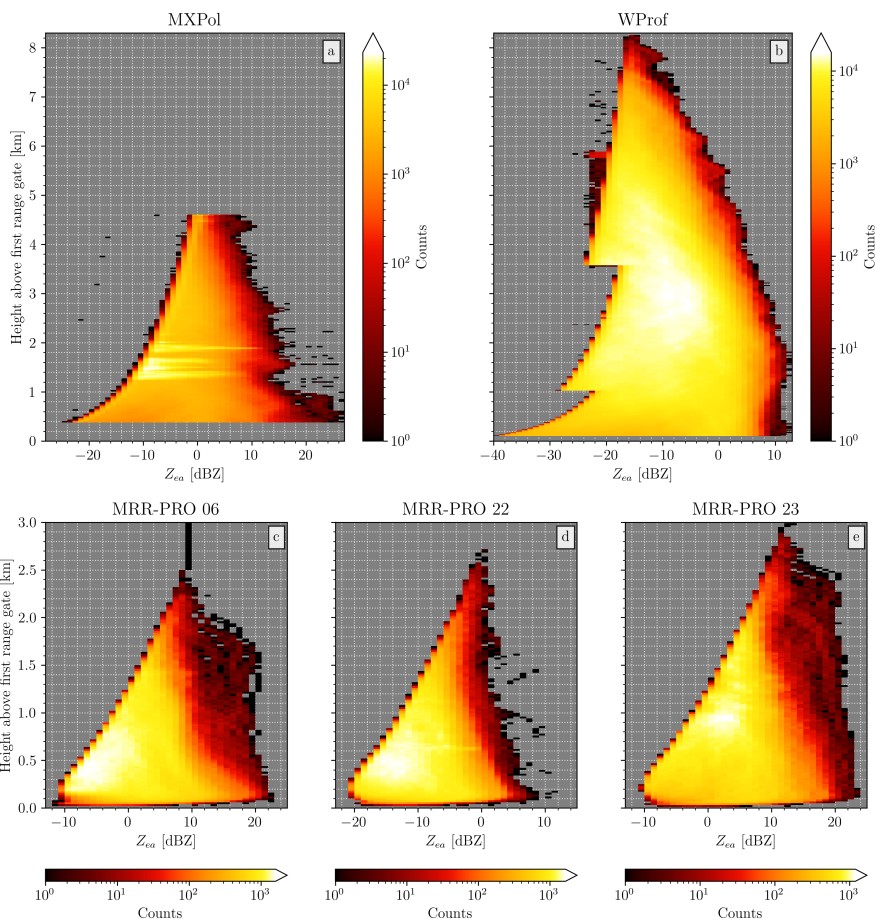

**Figure 4.** 2-dimensional histogram of the horizontal reflectivity factor collected by MXPol during the vertical scans (panel a), and the equivalent reflectivity factor recorded WProf (panel b), MRR-PRO 06 (panel c), MRR-PRO 22 (panel d), and MRR-PRO 23 (panel e). A minimum threshold of $0$ dB has been imposed on $SNR$ for all the radars.

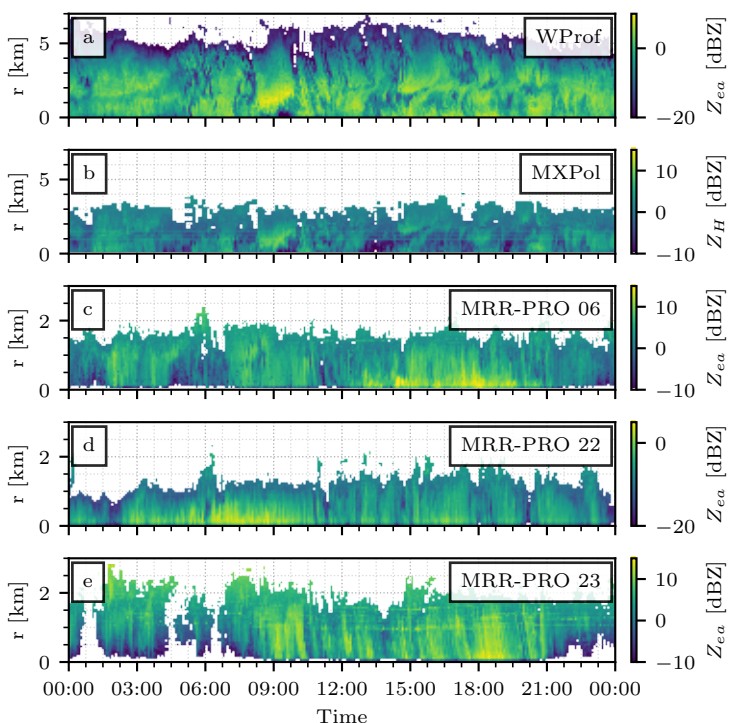

**Figure 5.** Measurements collected by the meteorological radars during the precipitation event on the 23 December 2019. The attenuated equivalent reflectivity factor collected by WProf and the three MRR-PRO is shown in panel a, c, d and e. Panel b shows the horizontal reflectivity factor recorded by MXPol during the vertical PPI scans. Conditions on $SNR$ analogous to the ones in Figure 4 have been enforced on all radars. An additional condition on the co-polar correlation coefficient ($\rho_{hv} > 0.6$) has been imposed on the MXPol measurements. Due to the limited sensitivity of the MRR-PRO, the vertical extent of the bottom three panels has been limited to the first 3 km of the profile.

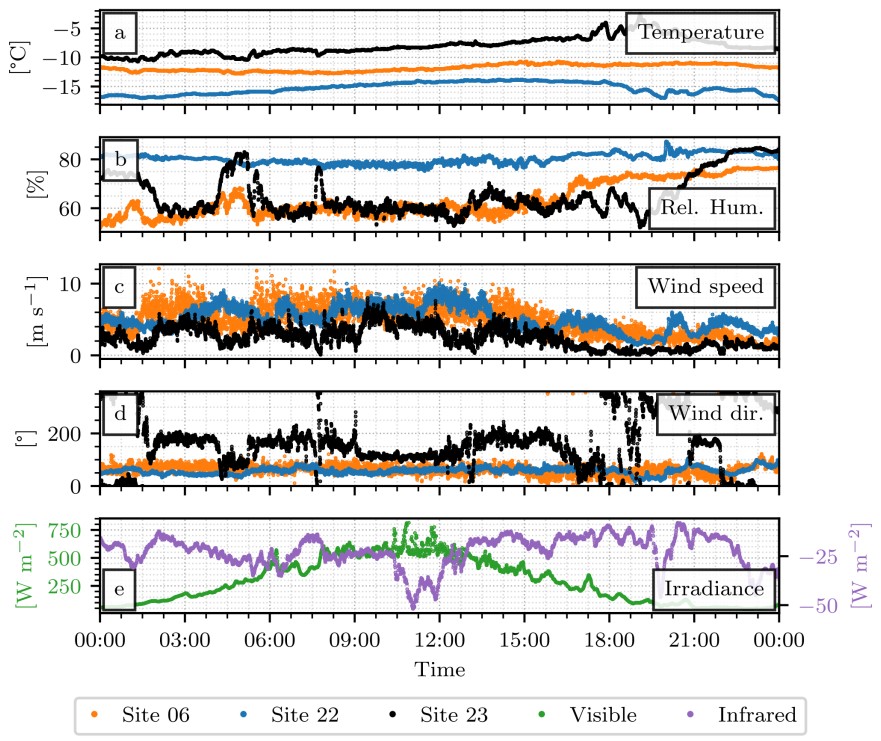

**Figure 6.** Measurements collected during the precipitation event on the 23 December 2019 by the automated weather stations and radiometers deployed alongside the three MRR-PRO. Four meteorological variables are displayed in the top four panels: temperature (panel a), relative humidity with respect to water (panel b), wind speed (panel c) and direction (panel d). The time-series at each panel is denoted by a different color: orange for the MRR-PRO 06 site, blue for the MRR-PRO 22 one, and black for the MRR-PRO 23 one. The downwelling irradiance in the shortwave (in green, y-axis on the left) and in the longwave (in purple, y-axis on the right) recorded by the radiometers co-located with the MRR-PRO 06 are shown in panel e.

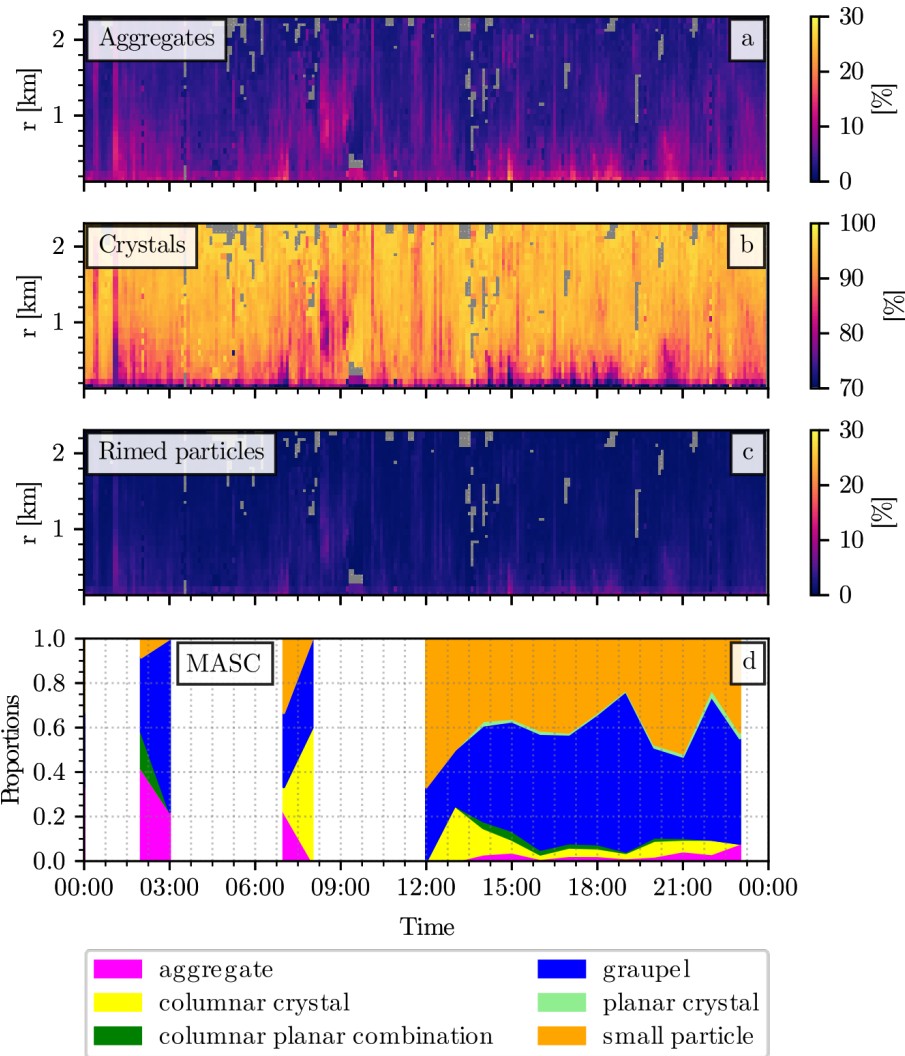

**Figure 7.** Information of the hydrometeor types derived from radar data and MASC images during the precipitation event on the 23 December 2019. The first three panels show the proportion of three hydrometeor classes computed using the algorithm described by Besic et al. (2018) from the RHI scans of MXPol performed above the base: snowflake aggregates in panel a, ice crystals in panel b and rimed particles in panel c. The hydrometeor types identified in the MASC images, using the method of Praz et al. (2017), are presented in panel d.

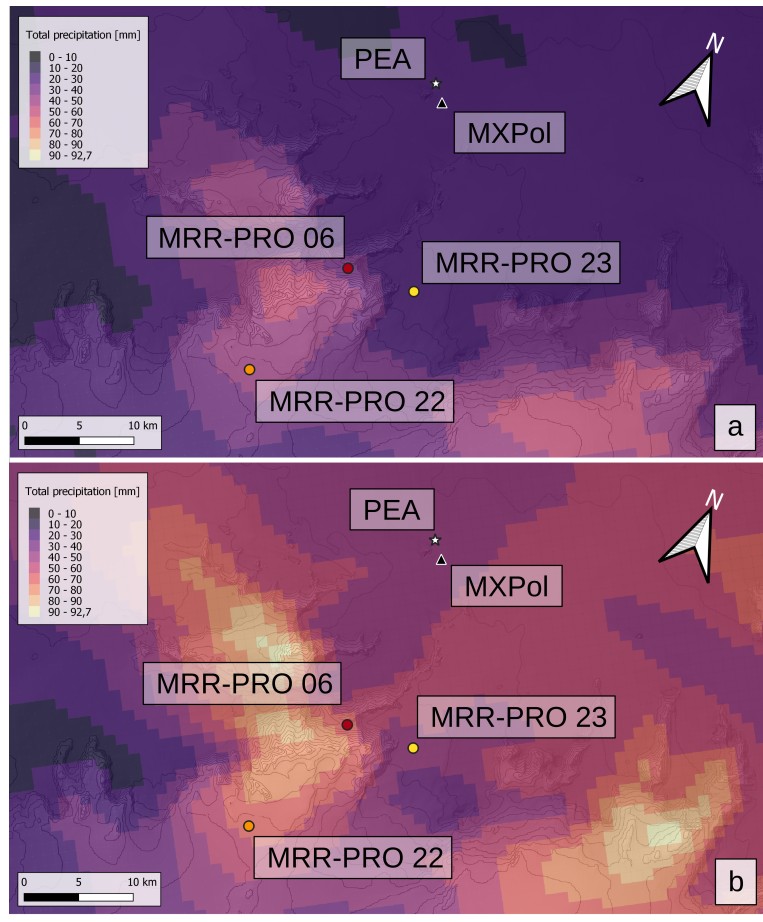

**Figure A1.** The total precipitation accumulation simulated by the WRF models in the innermost domain for December 2019 (panel a) and January 2020 (panel b), superimposed to a grayscale version of the digital elevation model already presented in Figure 1.