# Peer review of "Radar and ground-level measurements of clouds and precipitation collected during the POPE 2020 campaign at Princess Elisabeth Antarctica"

_Earth System Science Data, 2022_

## Author Comment (AC1)

**Reply to referee comments**

We thank the reviewers for the comments and suggestions, which helped clarify part of the manuscript and improve the quality of the datasets.

In the following, the comments from the reviewers are written in italic and highlighted in green, our replies in black, and the sentences added to the revised manuscript are highlighted in blue.

**1. Reply to RC 1**

**1.1 General remarks**

That said, I am disappointed that the authors decided not to share the radar spectra data in a repository, given that there are numerous free options for the storage of very large datasets open to the community. For example, even at the authors' home institute, there is a data repository enabling free uploads of up to 10TB (see https://www.epfl.ch/campus/library/acoua-support/), which I think should be sufficient for 2-months of radar spectra data, assuming repository volume constraints are the reason the spectra data were not uploaded, as noted in the Data Availability section.

We agree on the importance of sharing the Doppler spectra, but we were not aware of ACOUA before submitting the manuscript. We have now contacted the ACOUA support, and the spectra will be made available in an online repository. Unfortunately, it will not be possible to associate a DOI to this second repository, but the data storage will be guaranteed by ACOUA for a duration of 10 years (the maximum duration allowed in the standard procedure), and monthly checks will be performed to ensure that the data are not corrupted.

Aside from that, I am surprised that the authors submitted the manuscript in its current form with two non-readable figures(!): both figures 3 and 4 lack or contain bogus axis ticks, scales, and titles.

We apologize for the status of the figures. They always appeared with the correct labels, ticks, and titles while we were writing the manuscript on "overleaf" and in the pdf file we downloaded from the website. The file must have been corrupted during the upload, and we did not check the appearance of the figures after the upload.

We will control the quality of the figures after the upload of the revised manuscript.

**The text has quite a lot of typos and grammatical errors, some of which are listed below**

The typos listed have been corrected. In the next section, comments providing a correction to typos have not been listed, even though the suggested corrections have all been implemented.

**1.2 Other comments**

- *Table 1 is missing an explanation/definition of the different parameters.* An explanation of the parameters has been added in the caption of the table.
- This is the first time site coordinates are provided. I think that the station coordinates should be provided in the first instance mentioning the station in the main text. The coordinates of the station have been moved to line 17, where PEA is first mentioned.

- *here and elsewhere, nunatak is not capitalized, which is confusing.* The section describing the three MRR-PRO has been moved before the MXPol one so that they can be safely referenced when describing the scan strategy of MXPol.
- onward here the authors use 'latitude' and 'longitude' to specify coordinates, whereas previously they specified 'N' and 'E'.

Coordinates are now always preceded by the words "latitude" and "longitude", and the values are expressed in degrees, following a suggestion from the second reviewer.

- Doviak and Zrnic please provide a relevant chapter since this is a pretty long textbook. We added the relevant chapter (#8) in the reference.
- 3.1.2 is there an estimate of the magnitude of potential calibration offset/drift from July 2018 until the actual deployment date more than a year later?

Unfortunately, we do not have such an estimate. The time available to prepare the radar before the measurement campaign was relatively short, since the radar was deployed at another Antarctic station (Davis) in the austral summer preceding the measurement campaign at PEA, and it had to be shipped back from Australia and undergo some intervention to strengthen the box that contains its electronics.

The lack of information about a potential calibration drift has been included in the manuscript.

- sentence reads awkwardly recommend rewording. The sentence has been rewritten as: The raw Doppler spectra collected by WProf were not kept in their entirety: only the power identified as signal according to the radar manufacturer's algorithm (Küchler et al., 2017) has been saved.
- non-meteorological returns can have SNR much greater than 0 dB, so the question here is what do the authors mean in the text?

The reasoning behind the threshold has been clarified in the manuscript: While this condition alone does not guarantee the removal of all non-meteorological returns, it allows us to filter out faint peaks in the Doppler spectra that could have mistakenly been detected as precipitation.

• Since this is an article that describes the full radar dataset, I would like to see the comparison between the three radar types without being required to search for a different article. In its current form, I cannot evaluate the rest of this paragraph.

The 2-dimensional histograms for the equivalent reflectivity factor collected by the three MRR-PRO have been included in the figure.

• *if graupel is common yet there's a lack of rimed particles in the MXPol data, there is an inconsistency between the MASC and the MXPol. This is further emphasized in fig. 6, where there is an inconsistency in the timing of relatively higher riming occurrence. Since MASC directly captures particle images, I presume that its classification is more robust than a remote-sensing retrieval. So the question asked is how accurate and what is the value of the MXPol particle classification retrieval? How can these retrievals be used without reaching questionable conclusions? Guidance must be provided to users concerning the limitations of those retrievals*

We agree on the likelihood of the MASC being the instrument providing the most reliable hydrometeor classification. The classification (and detection of the proportion of the various hydrometeors in each volume, in short "demixing") provided by MXPol may in some cases not reflect the true hydrometeor mixture in the atmosphere. However, the two radar-based methods have been successfully used with MXPol data in the past, such as in the study of the microphysics and dynamics of snowfall in Korea by Gehring et al. 2019 (DOI: https://doi.org/10.5194/acp-20-7373-2020) and in the study of an atmospheric river event at Davis, Antarctica, by Gehring et al. 2022 (<a href="https://doi.org/10.1029/2021JD035210">https://doi.org/10.1029/2021JD035210</a> ).

Therefore, we consider the information added by the radar-based two methods a useful addition to the polarimetric variables already included in the file.

As suggested by the reviewer, our previous version of the manuscript lacked a clear statement on the limitations of these retrievals. Therefore, the section in the manuscript describing the hydrometeor classification and demixing has been expanded to address the issue of its reliability:

The same methods and centroids used to estimate the hydrometeor mixtures and dominant hydrometeor types during POPE have already been applied to measurements collected in another measurement campaign in Antarctica, albeit at a coastal location rather than an inland one (Gehring et al., 2022).

In our case, an accurate evaluation of the reliability of these two estimates cannot be provided for the current dataset.

Some discrepancies can be observed between their value and the one determined by the MASC at the ground level, as exemplified in Section 4.

Therefore, the two estimates should be independently verified through the observation, in the polarimetric variables, of the microphysical signatures associated with each hydrometeor type, before using them in scientific analyses.

• I think it is deceptive to claim that (useful) data were collected from November since only the Wband radar was operated towards the end of November and this was also in a test/calibration mode in the first several days as noted by the authors, so the effective date range should be December to February

The range has been corrected, indicating December as the start date.

- *missing reference for ERA5* The reference has been added
- *I would add text to the central panel specifying the location of the different sites mentioned by the authors (verheyefjellet, nunatak, etc.)* The text for the landmarks mentioned in the manuscript (the Nunatak Utsteinen, Vikinghogda, and the Verheyefjellet) has been added to the central panel.
- Still Fig. 1 I cannot differentiate between the red and brown lines. Also, I see dasheddotted and dotted but not a dashed line as specified in the caption. Also, I recommend shading the azimuth range not covered in the PPI scans, as indicated in the text. The brown line has been replaced by a yellow one. We apologize for the mistake in the caption, "dashed" has been now replaced by "dotted", when referring to the red lines showing the two RHI scans in the second cycle. The azimuth range excluded from the PPI scan has been shaded in gray.
- *for end users, it would be useful to provide the actual date on the x-axis.* The date has been added to the x-axis
- downwelling IR cannot be evaluated because of the different magnitudes relative to the SW. Recommend plotting on a different scale. In the revised version, the SW has been plotted on a different scale, presented on the y-axis on the right side of the panel.

**2. Reply to RC 2**

**2.1 General remarks**

- *Yet the manuscript lacks a section dedicated simply to presentation of the retrieved data variables* We thank the reviewer for the remark, an appendix summarizing the data products has been included in the manuscript.
- does not describe in the manuscript an instrument which is present in the online data archive, We do not understand which instrument the reviewer is referring to in this comment. All the instruments in the online repository have been presented in the manuscript:
  - WProf in section 2.1.1
  - MASC in section 2.1.2
  - MXPol in section 2.2 of the first version of the manuscript (section 2.3 of the revised version)
  - MRR-PRO in section 2.3 of the first version of the manuscript (section 2.2 of the revised one)
     Automated weather stations (AWS) and radiometers in section 2.3.1

Following the suggestion of the reviewer, we expanded the explanation on the AWS deployment. We are ready and willing to include additional explanations in future iterations, in case we missed any important information.

• and suffers from some editing errors in the text The errors in the text mentioned in the comments have been corrected.

**2.2 Major comments**

section 2.2: The online listing of the data states `This archive contains the radar variables collected by the W-band Doppler profiling cloud radar (WProf) deployed at PEA. The liquid water path and integrated water vapor (retrieved thanks to the 89 GHz radiometer included in the instrument) has also been included in the files,' but there is no mention of the LWP or IWV retrievals in the paper. The retrieval was mentioned in the original manuscript, in section 3.2, between lines 243 and 245: "Additionally, WProf includes an 89 GHz radiometer, which allowed the retrieval of the liquid water path and integrated water vapor using the algorithm described by Billault-Roux and Berne (2021)". In the revised version we included the acronym for both quantities to improve clarity.

section 2.2: Two different types of scan cycles are defined for MXPol, though there's no statement of how long each scan cycle takes and when they were performed over the measurement campaign. How long does each cycle take, and how were the two cycles used across the entirety of the measurement campaign? If they were switched at the investigators' discretion, a plot of which scan cycle the MXPol was in for the duration of the field campaign would be useful, or at least a quantification of how frequently each of the scan cycles was used.

The duration of each scan cycle has been added to their description.

We agree with the reviewer on the need for a clear visualization of the change in scan strategy. Therefore, we included a figure (Figure 2 in the revised manuscript) that illustrates the period in which the different scan cycles have been used.

• section 3, 3.1, 3.2: The suite of instruments used is complex, and discussion of the output variables is intermingled with discussion of processing steps. The paper could benefit from the addition of a `dataset' section succinctly summarizing the output file variables and their organization. For the hydrometeor classification, the types are output variables are not specified. The Zenodo listing states that information about the proportion of different hydrometeor types is also calculated, but this is not mentioned in the paper.

We thank the reviewer for the suggestion, a summary of the output variables has been included in the appendix. The summary also includes the output of the hydrometeor classification.

The information on the hydrometeor proportion was already included in the manuscript, in section 3.1.4, in lines 237-238:

"The same scans have been used to determine the hydrometeor mixture in the radar volumes, using the method described by Besic et al. (2018)".

Following the suggestion of the first reviewer, the section on the hydrometeor classification and demixing has been expanded in the revised version.

- *figure 3: Panel labels (a,b) are missing, as are axis labels, axis tick labels, colorbar tick labels, and colorbar labels. This probably occurred from a formatting issue during typesetting. fig. 4: This figure has the same issues as figure 3. When I open the PDF file, I see missing labels, missing text, and incorrectly formatted tick labels*We are sorry for this issue. As mentioned in the reply to the first reviewer, we did not check the correct appearance of the figure after the submission. We will perform the check when uploading the revised manuscript. *dataset: I spot-checked the MRR-Pro and MXPol\_PPI archives in Python with Xarray for data*
- *quality.* Not all variables contain a `long\_name' or a `standard\_name' attribute. For example, the MRR-PRO data variables called `Zea', `width', etc. only contain the `units' attribute. Some MXPol variables do not contain any attributes.

We thank the reviewer for checking the dataset and pointing out the lack of some metadata. We have now re-generated the attributes of the files and the metadata of the variables. Unfortunately, the "standard\_name" is based on the CF Standard Name Table (https://cfconventions.org/Data/cf-standard-names/current/build/cf-standard-name-table.html), and not all radar variables have a standard\_name associated with them, therefore we could not add this field to all variables. However, a long\_name has been added to each of them.

• dataset: When I open the MRR-PRO files with Xarray in Python, I see that valid radar profiles use NaNs to indicate range bins with no meteorological signal, but there is no missing value attribute or data quality variable. Are all processed profiles valid, and NaNs are simply used to indicate no meteorological signal detected? Or do NaNs indicate both `no signal' and `suspect data'? If the latter is the case, then additional information should be provided.

The second interpretation is the correct one: NaNs indicate both `no signal' and `suspect data'. Additional information has been included in the description of the dataset.

**2.3 Minor Comments**

profile.

- *line 52: when you say `The relatively high number of studies that were enabled by the availability of this dataset', it seems you are not referring to a specific dataset (such as the one presented in this study), but rather to the availability of the data from the instruments at the research base. If you are referring to a specific dataset, consider providing a link to it or citing it to avoid confusion.* We apologize for the lack of clarity in the text. The HYDRANT dataset is now mentioned explicitly in the text.
- lines 90-92: Why was this chirp table chosen? The text could benefit from elaborating on how the three resolutions benefit the measurements and what they are targeted for.
   Additional information has been included in this section:
   This particular split allows a minimum detectable reflectivity factor below -15 dBZ throughout the profile. As sensitivity decreases with height, the transition to the region of the profile associated to the next chirp brings a sharp improvement in sensitivity, while the vertical resolution and the Nyquist velocity interval decrease. Therefore, the measurements are characterized by a relatively high vertical resolution in the lowest range gates, and the resolution is progressively lowered to keep a sufficiently high sensitivity for the whole profile. The decrease in Nyquist velocity in the second and third chirp reflects the lower expected vertical velocity expected in the upper regions of the
- table 1: What is \$v\_{ny}\$, and does `Vel. res.' stand for velocity resolution? Please specify more information about the variable names in the table caption.
   The information has been added to the caption.

 lines 125-133: The outline of the scanning modes could benefit from easier comparison between the list of the scans used and the lines on figure 1 -- I would suggest either adding a labeled grid to the plot indicating the direction of 0\$^\circ\$, 90\$^\circ\$, 270\$^\circ\$ azimuth, and/or indicating the color/linestyle of the line used in fig. 1 within the list in the text. Additionally, the red and brown coloring is hard to tell apart.

Two white lines indicating the 0° and 90° azimuth have been added to the figure. The brown color has been replaced by yellow.

- lines 137-138: I would suggest reminding the reader of the azimuth angles of the RHI scans directed towards the MRR-PRO sites (i.e. 165.6\$^\circ\$ and \$190.1^\circ\$). This measurement cycle is complex and the paper would benefit from attempting to further disambiguate its presentation. The values of the two angles have been included in the text.
- lines 165-173: Specify longitude and latitude with \$^\circ{\mathrm{S}}\$ and \$^\circ{\mathrm{E}}\$ rather than using negative numbers. Latitude and longitude have been all converted to the suggested format.
- lines 175-176: While temporal coverage for each instrument is mentioned in the instrument's respective section, I think the paper would benefit from a plot indicating the period of coverage for each instrument so that the reader can get a better sense of the temporal overlap between instruments.
   The period of observation of each radar has been included in Figure 2, which also shows the

The period of observation of each radar has been included in Figure 2, which also shows the different scan cycles used by MXPol, and the change in chirp table of WProf.

- line 178: You specify two AWS instrument models but do not specify the difference are different MRR-PROs accompanied by different instrument models? If so it may be worthwhile to state the difference between the two models, or if they produce the same results with the same resolution etc. The section has been re-written, including the additional information:
   An automated weather station, manufactured by Vaisala, has been installed in the vicinity of each of the MRR-PRO, at a height of approximately 1.5 m above the ground. Two different models of weather stations have been used: the models WXT536, co-located with the MRR-PRO 22 and MRR-PRO 23, and the WXT520, at the MRR-PRO 06 site. The weather stations use the same measurement setup, their data have been collected at the same temporal resolution, and we do not expect any significant discrepancy because of the difference in the model. The values of wind direction and speed, atmospheric pressure, air temperature, and relative humidity with respect to liquid water have been collected ensuring time synchronization with the radar measurements.
- section 3.4: Please clarify whether figure 3 is a joint histogram, or a joint PDF. If it's a joint PDF, then referring to an area of frequent occurrence as `counts' may not be accurate. Review of this section is hindered by the formatting issues with figure 3.
   Figure 3 (figure 4 in the revised manuscript) is a joint histogram, the word histogram has now been mentioned explicitly in the caption of the figure and in the main text of the article.
- *line 265: Does figure 3 plot the joint PDF over the entire measurement campaign? This would be worth noting,*

We are sorry, but we do not understand this comment.

The joint histogram in Figure 3 (Figure 4 in the revised version) contains data from the whole campaign. The aim of this figure is to help visualize the sensitivity of the three instruments and its dependence on range. All the scans have been used to ensure that even the faintest signals recorded during the measurement campaign are represented in the histogram. The figure follows a similar analysis to the one presented in Gehring et al., 2021 (https://doi.org/10.5194/essd-13-417-2021) for the same radars during the ICEPOP 2018 measurement campaign.

• line 276-7: `By comparing the three datasets, a significant difference between the MRRPRO 22 curve and the ones from the other two MRR-PRO can be noticed.' What is the curve in question? The reader cannot `notice' a difference since it refers to a figure not included in this paper. Perhaps the authors could instead state that another paper found that the MRR-PRO 22 has a significant bias

**compared to the other two MRR-PRO instruments.**

The 2-dimensional histogram of the equivalent reflectivity factor collected by the three MRR-PRO has been included in Figure 4, alongside the ones for MXPol and WProf.

The discussion has been re-phrased to avoid direct references to figures in other articles and mentions to the "curves" computed in the Ferrone et al, 2022 article have been removed. The section has been rephrased as:

The 2-dimensional histograms computed for the MRR-PRO 06, 22, and 23 are shown in Figure 4.c, 4.d, and 4.e, respectively. By comparing the three datasets, a significant difference between the range of Zea values recorded by the MRR-PRO 22 and the ones from the other two MRR-PRO can be noticed.

• line 280-281: This discussion is unclear -- please state why comparing the lowest 1% of the data is useful for estimating the bias of the instrument. The implication seems to be that this quantity ought to be the same across all three instruments, but the authors should state why.

The comparison was designed to provide a rough estimate of the offset between the MRR-PRO 22 and the other two MRR-PRO, by comparing their sensitivity (with range) curves.

Since the three radars have the same hardware, we do not expect drastic differences between the faintest reflectivity factor that can be recorded by each system throughout the profile. Therefore, by comparing the quantile 0.01 of the empirical distributions of the MRR-PRO 22 and the other two MRR-PRO at each range gate, we can compute an approximate offset. While the offset is not precise enough for quantitative comparisons between the three profilers, it could be useful for qualitative comparisons of their measurements.

In the revised manuscript, the paragraph has been rewritten as:

Since the three MRR-PRO share the same hardware specifications, we expect a similar sensitivity in their measurements. Following the analysis presented in Ferrone et al. (2022), we can use the quantile 0.01 of the empirical Zea distribution at each range gate to estimate the sensitivity. The difference between the quantile 0.01 of the MRR-PRO 06 distributions and the same quantile in the MRR-PRO 22 ones is 9 dB, and the interquartile range (quantile 0.75 minus quantile 0.25) of this difference is equal to 2 dB. The difference between the quantiles 0.01 of the MRR-PRO 23 and MRR-PRO 22 is 11 dB, and the interquartile range of this difference is also 2 dB. The similarity between the two differences suggests that an offset of 10 dB should be added to the MRR-PRO 22 with an adequate reference set of measurements and the approximate nature of the offset estimate provided above, its value should not be used for quantitative comparison between the MRR-PRO 22 and the other two MRR-PRO. The 10 dB have not been added to the MRR-PRO 22 measurements provided in the publicly accessible datasets.

- *line 281:* `For both sets of differences, the interquartile range is 2 dB.' Previously you simply compare the threshold of the 1st percentile, so what is this the interquartile range of?
   The interquartile range has been computed on all the differences between 2 given MRR-PRO, to highlight the relatively high variability of the difference throughout the profile. A new explanation has been written in the revised version (provided in the previous answer).
- *line 290: This statement about valid data seems to suggest outages between the start and end date of observations, but these are not mentioned in the paper or the dataset landing page.* The term "valid" is indeed redundant and misleading, and therefore it has been removed in the revised manuscript. The time series provided in the new Figure 3 should help visualize the time of operation of the five radars.
- fig. 5: This figure does not have any of the same errors as the previous two. However, in panel (e), the longwave downwelling flux is so small so as to be unreadable and to appear to be negative for much of the time. This could be fixed with a secondary y-axis or separate panels for LW and SW downwelling irradiance.

In the revised version, the SW has been plotted on a different scale, presented on the y-axis on the right side of the panel.

---

## Author Response (AR1)

**Point-by-point reply to the reviews**

We thank once again the reviewers for their invaluable input, which allowed us to improve the quality of the manuscript and of the datasets.
Similarly to the previous reply, the comments from the reviewers are written in italic and highlighted in green, our replies in black, and the sentences added to the revised manuscript are highlighted in blue.
The line numbers in the track change file, hereafter "latexdiff", will be used in all our replies.

**1. Reply to RC 1**

**1.1 General remarks**

*That said, I am disappointed that the authors decided not to share the radar spectra data in a repository, given that there are numerous free options for the storage of very large datasets open to the community. For example, even at the authors' home institute, there is a data repository enabling free uploads of up to 10TB (see https://www.epfl.ch/campus/library/acoua-support/), which I think should be sufficient for 2-months of radar spectra data, assuming repository volume constraints are the reason the spectra data were not uploaded, as noted in the Data Availability section.*

As mentioned in the previous reply during the interactive discussion, at the time of the submission of the article we were not aware of ACOUA and we could not find a reliable online storage for all the Doppler spectra, given their relatively large size.

The Doppler spectra collected by the three MRR-PRO are stored in relatively small file, resulting in three archives with a total size of approximately 10 GB. Therefore, they could be uploaded on Zenodo, at the following link:
https://doi.org/10.5281/zenodo.7507087 .

The Doppler spectra collected by MXPol and WProf, instead, result in a total archive size of approximately 500 GB, even after using the maximum compression factor in the creation of the netCDF4 files.
Following the suggestion of the reviewer, the spectra of these two radars have been uploaded on ACOUA.
The ACOUA platform is relatively recent, and the setup of the procedure for uploading and sharing our particularly large dataset required more than two months.
The data can now be accessed can be requested through the following Infoscience entry:
https://infoscience.epfl.ch/record/299685?ln=en
A DOI has been assigned to the Infoscience entry: 10.5075/epfl-lte-299685 .
ACOUA guarantees the storage of the files for 10 years, and monthly checks on the data quality of the files will be performed to ensure that the data are not corrupted.

Unfortunately, ACOUA currently does not allow the possibility to associate a DOI with a size of 500GB, such as ours. Therefore, it was only possible to associate a DOI to the Infoscience entry, through which the dataset itself can be accessed.

*Aside from that, I am surprised that the authors submitted the manuscript in its current form with two non-readable figures(!): both figures 3 and 4 lack or contain bogus axis ticks, scales, and titles.*

We apologize for the status of the figures. They always appeared with the correct labels, ticks, and titles while we were writing the manuscript on "overleaf" and in the pdf file we downloaded from the website.
The file must have been corrupted during the submission, and we did not check the appearance of the figures after the upload.
We will carefully control the quality of the figures after the upload of the revised manuscript.

*The text has quite a lot of typos and grammatical errors, some of which are listed below*
The typos listed have been corrected.

**1.2 Other comments**

- *5 - remove 's' from 'profilers'*
  The 's' has been removed (line 5 of the latexdiff file)..

- *23 - missing space*
  The space has been added (line 23 of the latexdiff file)..

- *30 - the the*
  The duplicated "the" has been removed (line 31 of the latexdiff file).

- *61 - took part in the campaign*
  The word "to" has been replaced by "in" (line 62 of the latexdiff file).

- *62 - 'such as'*
  The word "such" has been added (line 63 of the latexdiff file).

- *81 - following what?*
  The word "following" has been replaced by "remaining sections of this article" (lines 75-76 of the latexdiff file).

- *Table 1 is missing an explanation/definition of the different parameters.*
  An explanation of the parameters has been added in the caption of the table.

- *99 - This is the first time site coordinates are provided. I think that the station coordinates should be provided in the first instance mentioning the station in the main text.*
  The coordinates of the station have been moved to line 17, where PEA is first mentioned.

- *120 - here and elsewhere, nunatak is not capitalized, which is confusing.*
  The word "Nunatak" has been capitalized in all the text (lines 18, 167, 184 of the latexdiff file).

- *125 - what is the MRR-PRO? Only the MRR-2 was mentioned thus far.*
  The section describing the three MRR-PRO has been moved before the MXPol one (to lines 117-153), so that they can be safely referenced when describing the scan strategy of MXPol.

- *165 - onward - here the authors use 'latitude' and 'longitude' to specify coordinates, whereas previously they specified 'N' and 'E'.*
  Coordinates are now always preceded by the words "latitude" and "longitude" (lines 17, 127, 130, 134-135 of the latexdiff file), and the value has been expressed in degrees, following a suggestion from the second reviewer.

- *195 - Doviak and Zrnic - please provide a relevant chapter since this is a pretty long textbook.*
  Chapter 8 has been specified in the reference (line 265)

- *212 - redundant 'a'*
  The redundant "a" has been removed (line 293 of the latexdiff file).

- *220 - missing reference and/or year.*
  The reference has been removed (line 293 of the latexdiff file), since in our opinion it was redundant. It was the URL of the page that described the radar target simulator on the website of the company "Palindrome". However, since the name of the company and the radar target simulator are explicitly mentioned the article, we decided to remove the link to the website.

- *3.1.2 - is there an estimate of the magnitude of potential calibration offset/drift from July 2018 until the actual deployment date more than a year later?*
  Unfortunately we do not have such an estimate. The time available to prepare the radar before the measurement campaign was relatively short, since the radar was deployed at another Antarctic station (Davis) in the austral summer preceding the measurement campaign at PEA, and it had to be shipped back from Australia and undergo some intervention to strengthen the box that contains its electronics.
  Mention of the lack of information about a possible drift in the calibration has been included in the manuscript (line 295 of the latexdiff file).

- *240 - sentence reads awkwardly - recommend rewording.*
  The sentence (lines 318-320 of the latexdiff file) has been rewritten as:
  The raw Doppler spectra were not saved, only the part above the noise level identified by the manufacturer's algorithm (Küchler et al., 2017) was.

- *249 - radiosoundings*
  The phrasing has been changed to "radiosondes launched" (line 330 of the latexdiff file).

- *250 - below then*
  The word "than" has been removed (line 332 of the latexdiff file).

- *265 - non-meteorological returns can have SNR much greater than 0 dB, so the question here is what do the authors mean in the text?*
  The reasoning behind the threshold has been clarified in the manuscript (lines 347-349 of the latexdiff file):
  While this condition alone does not guarantee the removal of all non-meteorological returns, it allows us to filter out faint peaks in the Doppler spectra that could have mistakenly be identified as precipitation.

- *269 - here and elsewhere: panel 3.a - this is confusing. Simply state "fig. 3a"*
  The word "panel" has been replaced by "Figure" (lines 352, 357, 361 of the latexdiff file).

- *276 - Since this is an article that describes the full radar dataset, I would like to see the comparison between the three radar types without being required to search for a different article. In its current form, I cannot evaluate the rest of this paragraph.*
  The 2-dimensional histograms for the equivalent reflectivity factor collected by the three MRR-PRO have been included in the figure (panles "c", "d", and "e" of Figure 4).

- *296 - missing '.'*
  The "." has been added (line 387 of the latexdiff file).

- *322-323 - if graupel is common yet there's a lack of rimed particles in the MXPol data, there is an inconsistency between the MASC and the MXPol. This is further emphasized in fig. 6, where there is an inconsistency in the timing of relatively higher riming occurrence. Since MASC directly captures particle images, I presume that its classification is more robust than a remote-sensing retrieval. So the question asked is how accurate and what is the value of the MXPol particle classification retrieval? How can these retrievals be used without reaching questionable conclusions? Guidance must be provided to users concerning the limitations of those retrievals*
  We agree on the likelihood of the MASC being the instrument providing the most reliable hydrometeor classification. The classification (and detection of the proportion of the various hydrometeors in each volume, in short "demixing") provided by MXPol may in some cases not reflect the true hydrometeor mixture in the atmosphere. However, the volumes measured by the two instruments differ significantly: while the MASC has a detection area with a diameter of few centimeters, a single range gate of MXPol has a length of 75 m. This difference will always result in a lack of agreement between the hydrometeor mixture estimated by the two instruments.
  Moreover, the hydrometeor classification and demixing have been used on MXPol with success in the past, such as in the study of an atmospheric river event at Davis, Antarctica, by Gehring et al.

2022 ( https://doi.org/10.1029/2021JD035210 ). Therefore, we consider the information added by the hydrometeor classification and demixing a useful addition to the polarimetric variables already included in the files.

In the example of Figure 6, the proportion of rimed particles detected by MXPol is much lower than the one observed by the MASC. The timing of higher riming occurrence is, in our opinion, not inconsistent. Considering the period after 12:00, in which the MASC records pictures for an uninterrupted period of few hours, the proportion of riming/groupel begins with low values in both the MXPol and MASC time series. The proportion increases in both instruments, reaching a maximum at 15:00. Around 18:00, the proportion of rimed particles decreases in both time series. The most noticeable mismatch can be seen after 20:15, when the increase in the proportion rimed particles observed MXPol precedes the MASC one by approximately 1 hour and 30 minutes.

As suggested by the reviewer, our previous version of the manuscript lacked a clear statement on the limitations of these retrievals. Therefore, the section in the manuscript describing the hydrometeor classification and demixing has been expanded (lines 311-316 of the latexdiff file) to mention the issue of its reliability:

The same hydrometeor classification method has been applied to measurements collected in another measurement campaign in Antarctica, albeit at a coastal location rather than an inland one (Gehring et al., 2022).

An accurate evaluation of the reliability of these estimates cannot be provided for the current dataset. Some discrepancies can be observed between their value and the one determined by the MASC at the ground level, as exemplified in Section 4.

The difference in sampling volume and altitude of measurements between the two sensors could be partially responsible for these discrepancy.

We therefore suggest to use those hydrometeor information with care.

We also slightly expanded the comparison between MXPol and the MASC in section 4 (lines 415-417 of the latexdiff file), adding the sentence:

Although the timing in the occurrence of graupel in the MASC classification and of rimed particles in the MXPol classification is overall consistent, discrepancies can be observed between their respective magnitude and relative importance compared to crystals and aggregates.

- *326 - I think it is deceptive to claim that (useful) data were collected from November since only the W-band radar was operated towards the end of November and this was also in a test/calibration mode in the first several days as noted by the authors, so the effective date range should be December to February*
  The range has been corrected, indicating December as the start date (line 419 of the latexdiff file).

- *347 - February 2107*
  The year has been corrected (line 444 of the latexdiff file).

- *350 - missing reference for ERA5*
  The reference has been added (line 447 of the latexdiff file).

- *1 - I would add text to the central panel specifying the location of the different sites mentioned by the authors (verheyefjellet, nunatak, etc.)*
  The text for the landmarks mentioned in the manuscript (the Nunatak Utsteinen, Vikinghogda, and the Verheyefjellet) has been added to the central panel.

- *1 caption - visibility --> detectability*
  The caption has been corrected.

- *Still Fig. 1 - I cannot differentiate between the red and brown lines. Also, I see dasheddotted and dotted but not a dashed line as specified in the caption. Also, I recommend shading the azimuth range not covered in the PPI scans, as indicated in the text.*
  The brown line has been replaced by a yellow one. We apologize for the mistake in the caption,

"dashed" has been now replaced by "dotted", when referring to the red lines showing the two RHI scans in the second cycle. The azimuth range excluded from the PPI scan has been shaded in gray.

- *2 - for end users, it would be useful to provide the actual date on the x-axis.*
  The date has been added to a secondary x-axis, on top of the panel (Figure 3 in the revised manuscript).

- *5 - downwelling IR cannot be evaluated because of the different magnitudes relative to the SW. Recommend plotting on a different scale.*
  In the revised version, the SW has been plotted on a different scale, presented on the y-axis on the right side of the panel (Figure 6 in the revised manuscript).

**2. Reply to RC 2**

**2.1 General remarks**

- *Yet the manuscript lacks a section dedicated simply to presentation of the retrieved data variables*
  We thank the reviewer for the remark, an appendix summarizing the data products has been included in the manuscript (Appendix B, lines 460-538 of the latexdiff file).
  The appendix is referenced in the text in the data processing section of the article (lines 277, 325, 343 of the latexdiff files).

- does not describe in the manuscript an instrument which is present in the online data archive,
  We did not understand which instrument is the reviewer referring to in this comment.
  All the instruments in the online repository have been presented in the manuscript:
  - WProf in section 2.1.1
  - MASC in section 2.1.2
  - MXPol in section 2.2 of the first version of the manuscript (section 2.3 of the revised version)
  - MRR-PRO in section 2.3 of the first version of the manuscript (section 2.2 of the revised one)
  - Automated weather stations (AWS) and radiometers in section 2.3.1

  Under suggestion of the reviewer, we slightly expanded the explanation on the AWS deployment (line 145-147 of the latexdiff file). Additional information on the scan cycle of MXPol has also been added (lines 198-214), since the description of the temporary cycle used at the beginning of the campaign (and included in the online dataset) were missing from the article.
  We will be happy to include additional explanations in future iterations, in case we missed any important information.

- and suffers from some editing errors in the text
  The error in the text mentioned in the comments have been corrected.

**2.2 Major comments**

- *section 2.2: The online listing of the data states `This archive contains the radar variables collected by the W-band Doppler profiling cloud radar (WProf) deployed at PEA. The liquid water path and integrated water vapor (retrieved thanks to the 89 GHz radiometer included in the instrument) has also been included in the files,' but there is no mention of the LWP or IWV retrievals in the paper.*
  The retrieval was mentioned in the original manuscript, in section 3.2, between lines 243 and 245: "Additionally, WProf includes an 89 GHz radiometer, which allowed the retrieval of the liquid water path and integrated water vapor unsing the algorithm described by Billault-Roux and Berne (2021)".

  In the revised version we included the acronym for both quantities to improve clarity. A mention of the automated weather station included in the radar has also been added to the section.
  The revised version of the text (lines 322-325 of the latexdiff file) is:
  "Additionally, WProf includes an automated weather station, whose measurements have been included in the data files, and an 89 GHz radiometer, which allowed the retrieval of the Liquid Water

Path (LWP) and Integrated Water Vapor (IWV) using the algorithm described by Billault-Roux and Berne (2021)"

- *section 2.2: Two different types of scan cycles are defined for MXPol, though there's no statement of how long each scan cycle takes and when they were performed over the measurement campaign. How long does each cycle take, and how were the two cycles used across the entirety of the measurement campaign? If they were switched at the investigators' discretion, a plot of which scan cycle the MXPol was in for the duration of the field campaign would be useful, or at least a quantification of how frequently each of the scan cycles was used.*
The duration of each scan cycle has been added to their description (lines 171, 186, and 200 of the latexdiff file).
We agree with the reviewer on the need for a clear visualization of the change in scan strategy. Therefore, we included a figure (Figure 2 in the revised manuscript) that illustrates the period in which the different scan cycles have been used.

- *section 2.3: The pointing of MXPol is discussed in detail, but the pointing of the MRR-PROs is not mentioned. What is the elevation and azimuth angle of these instruments? Is their orientation fixed for the duration of their deployment? How was their orientation chosen?*
The reviewer is correct in pointing out the lack of information on the orientation of the MRR-PRO. The following text has been added to the section describing the transect of MRR-PRO (section 2.2 in the revised manuscript, lines 120-123 of the latexdiff file):
"Each of the three radars has been installed on a vertical pipe, inserted in the ice directly below it, and stabilized using guy wires to four anchor points at different angles, to avoid excessive vibration of the instrument.
The vertical pointing of the instrument was checked using the circular level included in the radars."

Since information on the orientation of WProf was also missing from the article, we added the following text to section 2.1.1 (lines 92-93 of the latexdiff file):
"The vertical pointing of the radar was checked using two perpendicular levels placed on the instrument."

- *section 3, 3.1, 3.2: The suite of instruments used is complex, and discussion of the output variables is intermingled with discussion of processing steps. The paper could benefit from the addition of a `dataset' section succinctly summarizing the output file variables and their organization. For the hydrometeor classification, the types are output variables are not specified. The Zenodo listing states that information about the proportion of different hydrometeor types is also calculated, but this is not mentioned in the paper.*
We thank the reviewer for the suggestion, a summary of the output variables has been included in the appendix. The summary also includes the output of the hydrometeor classification.
The information on the hydrometeor proportion was already included in the manuscript, in section 3.1.4, at the lines 237-238:
"The same scans have been used to determine the hydrometeor mixture in the radar volumes, using the method described by Besic et al. (2018)".
Under the suggestion of the first reviewer, the section on the hydrometeor classification and demixing has been slightly expanded in the revised version (lines 311-316).

- *figure 3: Panel labels (a,b) are missing, as are axis labels, axis tick labels, colorbar tick labels, and colorbar labels. This probably occurred from a formatting issue during typesetting.*
*fig. 4: This figure has the same issues as figure 3. When I open the PDF file, I see missing labels, missing text, and incorrectly formatted tick labels*
As mentioned in the reply to the first reviewer, we did not check the correct appearance of the figure after the submission. We are now aware of the issue, and the figures will be all checked immediately after the upload of the revised manuscript.

- *dataset: I spot-checked the MRR-Pro and MXPol_PPI archives in Python with Xarray for data quality. Not all variables contain a `long_name' or a `standard_name' attribute. For example, the MRR-PRO data variables called `Zea', `width', etc. only contain the `units' attribute. Some MXPol variables do not contain any attributes.*

  We thank the reviewer for checking the dataset and pointing out the lack of some metadata. We have now re-generated the attributes of the files and the metadata of the variables. Unfortunately, the "standard_name" is based on the CF Standard Name Table ( [https://cfconventions.org/Data/cf-standard-names/current/build/cf-standard-name-table.html](https://cfconventions.org/Data/cf-standard-names/current/build/cf-standard-name-table.html) ), and not all radar variables have a standard_name associated to them, therefore we could not add this field to all variables. However, a long_name has been added to each of them.

  Additionally, information on all the long names of the variables has been added in Appendix B (lines 453-531).

- *dataset: When I open the MRR-PRO files with Xarray in Python, I see that valid radar profiles use NaNs to indicate range bins with no meteorological signal, but there is no missing value attribute or data quality variable. Are all processed profiles valid, and NaNs are simply used to indicate no meteorological signal detected? Or do NaNs indicate both `no signal' and `suspect data'? If the latter is the case, then additional information should be provided.*

  The second interpretation is the correct one: NaNs indicate both `no signal' and `suspect data'. The number of "attributes" in the MRR-PRO netCDF4 files has been increase, to ease the understanding of their content.

  In particular, the following string has been included under the metadata "comment" of the files: "Not-a-number (nan) values denote the range gates containing no signal, either because no significant peak has been detected, or because the value has been removed by the post-processing (flagged as interference). The attributes listed in the ERUO_processing_parameters provide information on the setting used by the ERUO processing."

  Additional metadata in the files now specify the coordinates of the instrument, range and velocity resolution, number of range gates and velocity bins, and all the parameters used by the ERUO library for the processing and post-processing of the raw spectra (listed under the attribute "ERUO_processing_parameters").

**2.3 Minor Comments**

- *line 23: change `SMB(of)' to `SMB (of)' (missing a space)*

  The missing space has been added (line 24 of the latexdiff file).

- *line 27: change `suggest that in Queen Maud land few' to `suggest that in Queen Maud land a few' (missing article)*

  The article "a" has been added  (line 28 of the latexdiff file).

- *line 52: when you say `The relatively high number of studies that were enabled by the availability of this dataset', it seems you are not referring to a specific dataset (such as the one presented in this study), but rather to the availability of the data from the instruments at the research base. If you are referring to a specific dataset, consider providing a link to it or citing it to avoid confusion.*

  We apologize for the lack of clarity in the text. The HYDRANT dataset is now mentioned explicitly in the text (line 53 of the latexdiff file).

- *lines 90-92: Why was this chirp table chosen? The text could benefit from elaborating on how the three resolutions benefit the measurements and what they are targeted for.*

  Additional information has been included in this section (lines 96-101 of the latexdiff file):

  This particular split allows a minimum detectable reflectivity factor below -15 dBZ throughout the profile. As sensitivity decreases with height, the transition to the region of the profile associated with the next chirp brings a sharp increase in sensitivity, while the vertical resolution and the Nyquist

velocity interval decrease. Therefore, the measurements are characterized by a relatively high vertical resolution in the lowest range gates, and the resolution is progressively lowered to keep a sufficiently high sensitivity for the whole profile. The decrease in Nyquist velocity in the second and third chirp is consistent with the lower vertical velocity expected in the upper regions of the profile.

- *table 1: What is $v_{ny}$, and does `Vel. res.' stand for velocity resolution? Please specify more information about the variable names in the table caption.*
  The information has been added to the caption.

- *lines 125-133: The outline of the scanning modes could benefit from easier comparison between the list of the scans used and the lines on figure 1 -- I would suggest either adding a labeled grid to the plot indicating the direction of $0^\circ$, $90^\circ$, $270^\circ$ azimuth, and/or indicating the color/linestyle of the line used in fig. 1 within the list in the text. Additionally, the red and brown coloring is hard to tell apart.*
  Two white lines indicating the 0° and 90° azimuth have been added to the figure. The brown color has been replaced by yellow.

- *lines 137-138: I would suggest reminding the reader of the azimuth angles of the RHI scans directed towards the MRR-PRO sites (i.e. $165.6^\circ$ and $190.1^\circ$). This measurement cycle is complex and the paper would benefit from attempting to further disambiguate its presentation.*
  The value of two angles have been included in the text (line 187 of the latexdiff file).

- lines 165-173: Specify longitude and latitude with $^\circ{\mathrm{S}}$ and $^\circ{\mathrm{E}}$ rather than using negative numbers.
  Latitude and longitude have been all converted to the suggested format (lines 17, 127, 130, 134-135 of the latexdiff file). Additionally, coordinates are now always preceded by the words "latitude" and "longitude", as suggested by the first reviewer.

- *lines 175-176: While temporal coverage for each instrument is mentioned in the instrument's respective section, I think the paper would benefit from a plot indicating the period of coverage for each instrument so that the reader can get a better sense of the temporal overlap between instruments.*
  The period of observation of each radar has been included in Figure 2, which also shows the different scan cycles used by MXPol, and the change in chirp table of WProf.
  The Figure is mentioned in the text at lines 91, 140, and 216 of the latexdiff file.

- *line 178: You specify two AWS instrument models but do not specify the difference – are different MRR-PROs accompanied by different instrument models? If so it may be worthwhile to state the difference between the two models, or if they produce the same results with the same resolution etc.*
  The section has been re-written, including the additional information (lines 143-149 of the latexdiff file):
  An automated weather station, manufactured by Vaisala, has been collocated to each MRR-PRO, at a height of approximately 1.5 m above the ground. Two different models of weather stations have been used: the models WXT536, co-located with the MRR-PRO 22 and MRR-PRO23, and the WXT520, at the MRR-PRO~06 site. The weather stations use the same measurement setup, their data have been collected at the same temporal resolution, and we do not expect any significant discrepancy in their observations because of the difference in the model. The values of wind direction and speed, atmospheric pressure, air temperature, and relative humidity with respect to liquid water have been collected ensuring time synchronization with the radar measurements.

- *section 3.4: Please clarify whether figure 3 is a joint histogram, or a joint PDF. If it's a joint PDF, then referring to an area of frequent occurrence as `counts' may not be accurate. Review of this section is hindered by the formatting issues with figure 3.*
  Figure 3 (figure 4 in the revised manuscript) is a joint histogram, the word histogram has now been mentioned explicitly in the caption of the figure and in the main text of the article.

- *line 249: Typo: `radiosondes', not `radiosoungins'*
  The typo has been corrected (line 329 of the latexdiff file).

- *line 265: Does figure 3 plot the joint PDF over the entire measurement campaign? This would be worth noting,*
  We are sorry, but we did not understand this comment.
  The joint histogram in Figure 3 (Figure 4 in the revised version) contains data from the whole campaign. The aim of this figure is to help visualizing the sensitivity of the three instruments, and its dependence with range. All the scans have been used to ensure that even the faintest signals recorded during the measurement campaign are represented in the histogram. The figure follows a similar analysis to the one presented in Figure 8 of Dias Neto et al., 2019 (https://doi.org/10.5194/essd-11-845-2019) and in Figure 3 of Gehring et al., 2021 (https://doi.org/10.5194/essd-13-417-2021).

- *line 276-7: `By comparing the three datasets, a significant difference between the MRRPRO 22 curve and the ones from the other two MRR-PRO can be noticed.' What is the curve in question? The reader cannot `notice' a difference since it refers to a figure not included in this paper. Perhaps the authors could instead state that another paper found that the MRR-PRO 22 has a significant bias compared to the other two MRR-PRO instruments.*
  The 2-dimensional histogram of the equivalent reflectivity factor collected be the three MRR-PRO has been included in Figure 4, alongside the ones for MXPol and WProf.
  The discussion has been re-phrased to avoid direct references to figures in other articles, and mentions to the "curves" computed in the Ferrone et al, 2022 article have been removed.
  The section (lines 360-362) has been rephrased as:
  The 2-dimensional histograms computed for the MRR-PRO 06, 22, and 23 are shown in Figure 4.c, 4.d and 4.e, respectively. By comparing the three datasets, a significant difference between the range of Zea values recorded by the MRR-PRO 22 and the ones from the other two MRR-PRO can be noticed.

- line 280-281: This discussion is unclear -- please state why comparing the lowest 1% of the data is useful for estimating the bias of the instrument. The implication seems to be that this quantity ought to be the same across all three instruments, but the authors should state why.
  The comparison was designed to provide a rough estimate of the offset between the MRR-PRO~22 and the other two MRR-PRO, by comparing their sensitivity curve.
  Since the three radars have the same hardware, we do not expect drastic differences between the faintest reflectivity factor that can be recorded by each system throughout the profile. Therefore, by comparing the quantile 0.01 of the empirical distributions of the MRR-PRO~22 and the other two MRR-PRO at each range gate, we can compute an approximate offset. While the offset is not precise enough for quantitative comparisons between the three profilers, it could be useful for qualitative comparisons of their measurements.
  In the revised manuscript, the paragraph (lines 365-376) has been rewritten as:
  Since the three MRR-PRO share the same hardware specifications, we expect a similar sensitivity in their measurements. Following the analysis presented in Ferrone et al. (2022), we can use the quantile 0.01 of the empirical $Z_{ea}$ distribution at each range gate to estimate the sensitivity.
  The difference between the quantile 0.01 of the MRR-PRO 06 distributions and the same quantile in the MRR-PRO 22 ones is 9 dB, and the interquartile range (quantile 0.75 minus quantile 0.25) of this difference is equal to 2 dB. The difference between the quantiles 0.01 of the MRR-PRO~23 and MRR-PRO 22 is 11 dB, and the interquartile range of this difference is also 2 dB.
  The similarity between the two differences suggests that an offset of 10 dB should be added to the MRR-PRO 22 to compensate for its lower $Z_{ea}$ values. Given the impossibility of comparing the MRR-PRO 22 with an adequate reference set of measures and the approximate nature of the offset estimate provided above, we suggest to use this offset with care. The 10 dB have not been added to the MRR-PRO~22 measurements provided in the publicly accessible datasets.

- *line 281: `For both sets of differences, the interquartile range is 2 dB.' Previously you simply compare the threshold of the 1st percentile, so what is this the interquartile range of?*
  The interquartile range has been computed on the differences, to highlight the relatively high variability of the difference throughout the profile. The explanation (lines 365-376) has been written

in the revised version (provided in the previous answer).

- *line 290: This statement about valid data seems to suggest outages between the start and end date of observations, but these are not mentioned in the paper or the dataset landing page.*
  The term "valid" is indeed redundant and misleading, and therefore it has been removed in the revised manuscript (line 381 of the latexdiff file). The time series provided in the new Figure 3 should help to visualize the time of operation of the five radars.

- *line 296: Missing period at the end of the sentence `.'*
  The missing period has been added (line 387 of the latexdiff file).

- *fig. 5: This figure does not have any of the same errors as the previous two. However, in panel (e), the longwave downwelling flux is so small so as to be unreadable and to appear to be negative for much of the time. This could be fixed with a secondary y-axis or separate panels for LW and SW downwelling irradiance.*
  In the revised version, the SW has been plotted on a different scale, presented on the y-axis on the right side of the panel.